# Archaeobotanical Study of Tell Khamîs (Syria)

Javier Valera [1,*], Gonzalo Matilla-Seiquer [1], Concepción Obón [2] and Diego Rivera [3,*]

[1] Departamento de Prehistoria, Arqueología, Historia Antigua, Historia Medieval y Ciencias y Tecn. Historiográficas, Facultad de Letras, Campus de la Merced, Universidad de Murcia, C/Santo Cristo, 1, 30001 Murcia, Spain; gmatilla@um.es

[2] Departamento de Biología Aplicada, EPSO, Universidad Miguel Hernández de Elche, Carretera de Beniel km 3.2, 03312 Orihuela, Spain; cobon@umh.es

[3] Departamento de Biología Vegetal, Facultad de Biología, Universidad de Murcia, 30100 Murcia, Spain

* Correspondence: javier.valera@um.es (J.V.); drivera@um.es (D.R.)

**Abstract:** Tell Khamîs, an archaeological site of the Syrian region of Upper Jazeera, is 3 km east of the Euphrates and 31 km from Yarâblûs (ancient Karkemiš); archaeological excavations determined seven different levels (Early Bronze, Middle Bronze, Aramaic, Assyrian, Persian, Hellenistic, and Islamic). This study aims to identify plant remains recovered during the excavation of the site and to place them within the chronology of Tell Khamîs and in the context of the archaeobotanical evidence for the Upper Euphrates. A total of 88 sediment samples were collected, and seeds, recovered via flotation, were identified using optical and SEM microscopy. A total of 20,606 whole remains and 37,646 fragments belonging to 92 taxa and 35 plant families were identified. Seed lists were compared with those from other sites, and the results were analyzed using multivariate techniques. Barley is particularly noteworthy for the number of remains; this species was found in 49 different samples, notably, in a silo of about 4 m$^3$ in volume. Middle Bronze Age and Assyrian levels are the richest in remains; the site presents a maximum of documented activity in the Middle Bronze Age period, and the most interesting taxa from a paleoenvironmental and cultural point of view are represented by one or a few seeds.

**Keywords:** paleobotany; archaeology; agriculture; Near East; crops; paleoethnobotany

## 1. Introduction

Archaeobotanical research is concerned with identifying plant remains in archaeological contexts as a means of inferring the relationship of past human populations to the paleoenvironment and paleoeconomy. The Near East harbors an extremely rich amount of archaeobotanical material in its archaeological sites, as evidenced by the abundance of works that, since the mid-20th century, and especially since the 21st century, have focused their efforts on studying agriculture, ancient diets, fuels used, the impact of human activities on the plant landscape and, especially, the practices of domestication and exploitation of plants; this is in an effort to delimit the areas where they originate and their chronology [1].

The Eastern Mediterranean area and the Fertile Crescent have been home to many of the world's earliest civilizations and have seen their rise and collapse [2]. During the Bronze Age, agricultural production laid the foundation for the development and maintenance of these societies at the state level [3]. While we find a great deal of specialized literature on the processes of domestication, the emergence of crop plants, and the transition from a subsistence based on the exploitation of wild resources to one in which agriculture plays a predominant role [4], there is less information on the development of crops in the Bronze Age and later periods.

The Near East is ideal for archaeobotanical investigations throughout its human history given the degree of preservation and conservation of plant remains and the great availability of these, especially in the archaeological contexts of storage and refuse disposal during the

Bronze Age. For some time, the northern region of Syria was considered as a backward rural periphery compared to the urban development of the southern cities, but recent research has revealed impressive cultural, economic, and demographic changes that imply significant environmental impacts in this chronological horizon [5]. In turn, a network of cities that traded agricultural surpluses with each other has been evidenced, along with cities that could occupy an important religious position and behave as redistribution centers for grain and other foodstuffs in this area of Syria [4].

The interrelationship between a society and the environment, deduced through archaeobotany, can provide a fairly accurate picture of cultivation strategies, soil depletion, and deforestation. Plant taxa associated with crops and weeds usually represent the largest proportion within archaeobotanical investigations, which is why the specialized literature has not considered them suitable for environmental reconstruction [6]. While it is true that most plant macro-remains are derived mainly from the processing and storage of crops intended for human consumption and reflect a specific ecological range associated with cultivation strategies, seeds recovered from silos and amortized pits or from refuse deposits may reflect the wild flora of the site environment.

Currently, a semi-arid steppe covers much of northern Syria. The absence of large geographic barriers means that in summer, the subtropical high-pressure belt moves northward, leaving a dry and warm environment. Atlantic low-pressure systems arrive in winter, bringing most of the storm precipitation to this region of the Eastern Mediterranean [7]. Based on the evidence found at Soreq Cave [8], it appears that aridity may have reached its present value during the first third of the Middle Bronze Age, although most scholars agree on a trend towards the present climate since the mid-Holocene [4].

Most archaeological sites in Syria are located in an area with an annual isohyet of between 200 and 400 mm [9], which turned into an area very sensitive to any change in precipitation. In these semi-arid environments, continuous water deficits have been a limiting factor for the cultivated plant assemblage and agricultural decision-making, but also for wild plants where steppe taxa predominate. To all this, we can add the risk of practicing irrigation, with a high evaporation rate that can cause the salinization of cultivated soils [10].

The main objectives of this study start with determining the identity of the plant remains recovered during the excavation of the various levels of the site, and to place them within the chronology of Tell Khamîs and in the main context provided by the archaeobotanical evidence available for the Upper Euphrates and surrounding areas.

With the data obtained, our further objectives are to determine the relevance of the site throughout its occupation; the plants-related activities that could be carried out, especially those of an agricultural nature; and the novel aspects offered by the findings.

Based on the results obtained from the analysis and identification of plant macro-rests, this paper aims to present a critical evaluation of the Tell Khamîs seed assemblage, offering an interpretation of the cultivation strategies, the plant economy, and the relationship between plants and the human communities that inhabited this site from the Early Bronze Age to the Hellenistic period. In turn, this research aims to provide an inter-regional comparison of archaeobotanical evidence from similar sites, from which we can infer insights into land-use patterns and their evolution in this area of Syria.

We also aim, through the analysis of the current preferred habitats of the species and genera identified, to obtain an idea of the environment of the site; its evolution, if any, over the periods studied; and its position in the context of the macro-region of other published sites.

## 2. Materials and Methods

### 2.1. The Site

Tell Khamîs is an archaeological site located in northern Syria within the Syrian region of Upper Jazeera (36°41′56.6″ N 38°11′53.3″ E), originally 3 km east of the Euphrates and 31 km from Yarâblûs (ancient Karkemiš) and 37 km from Mambig (ancient Hierapolis)

(Figure 1). At 330 m above sea level, it is a small hill inscribed in a square with a base of 100 m and a maximum height of 8 m above the surrounding plain [11].

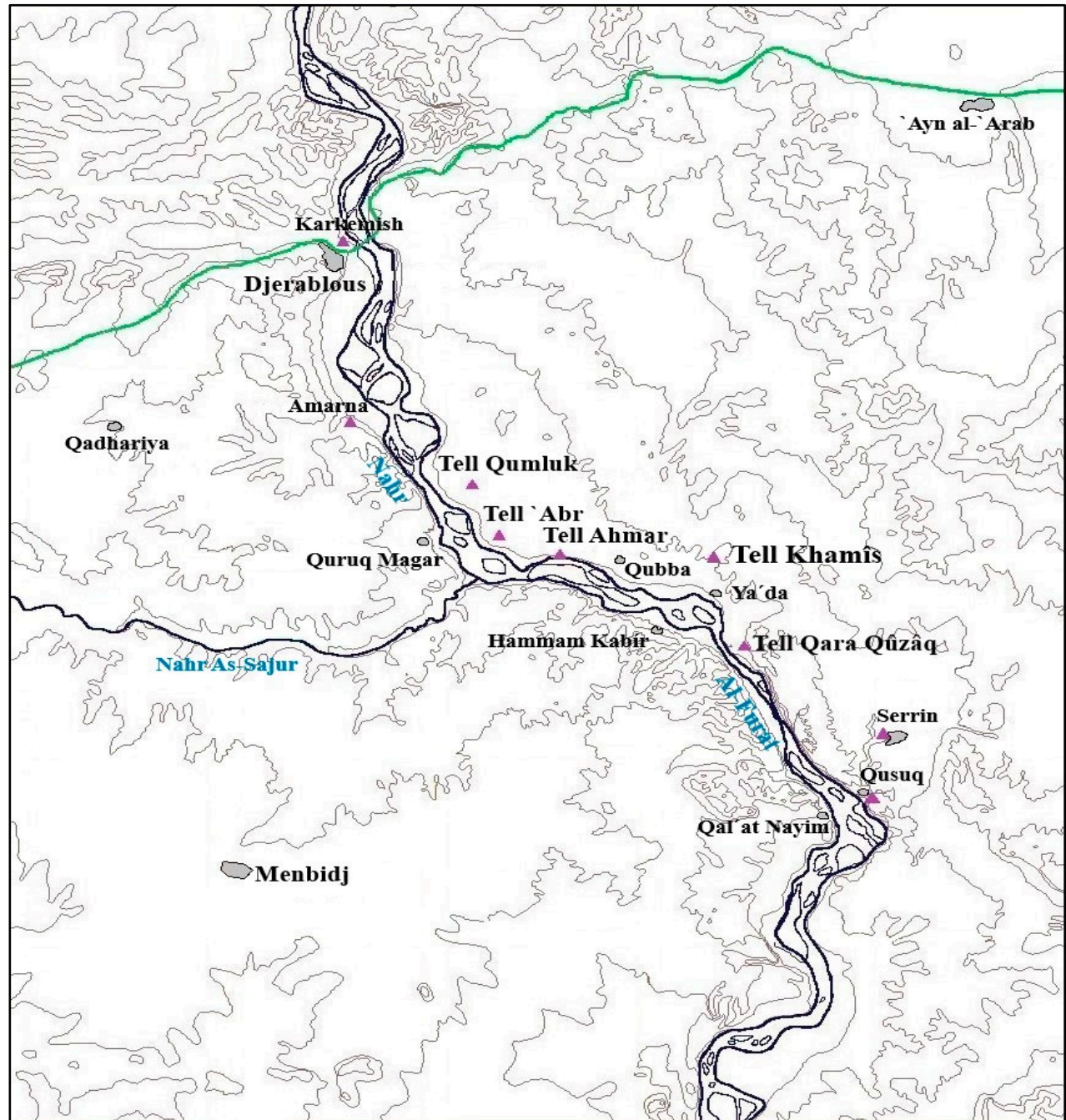

**Figure 1.** Location of Tell Khamîs in the context of the Upper Euphrates. Tell Khamîs is situated in the vicinity of Tell Qara Qûzâq by the banks of the Euphrates River. In green is the Syrian–Turkish borderline. Coordinates: UTM. Image: from Matilla [2] modified by Javier Valera, with permission.

Tell Khamîs was excavated by the University of Murcia between 1992 and 2000 within the archaeological rescue campaign caused by the construction of the Tishrin dam that affected a large number of sites [12]. The archaeological intervention made it possible to determine, in Tell Khamîs, seven different levels (Early Bronze II, Middle Bronze I, Aramaic, Assyrian, Persian, Hellenistic, and Islamic), some with great dynamism and without interruption, such as the Persian–Hellenistic, with five different phases (Figure 2).

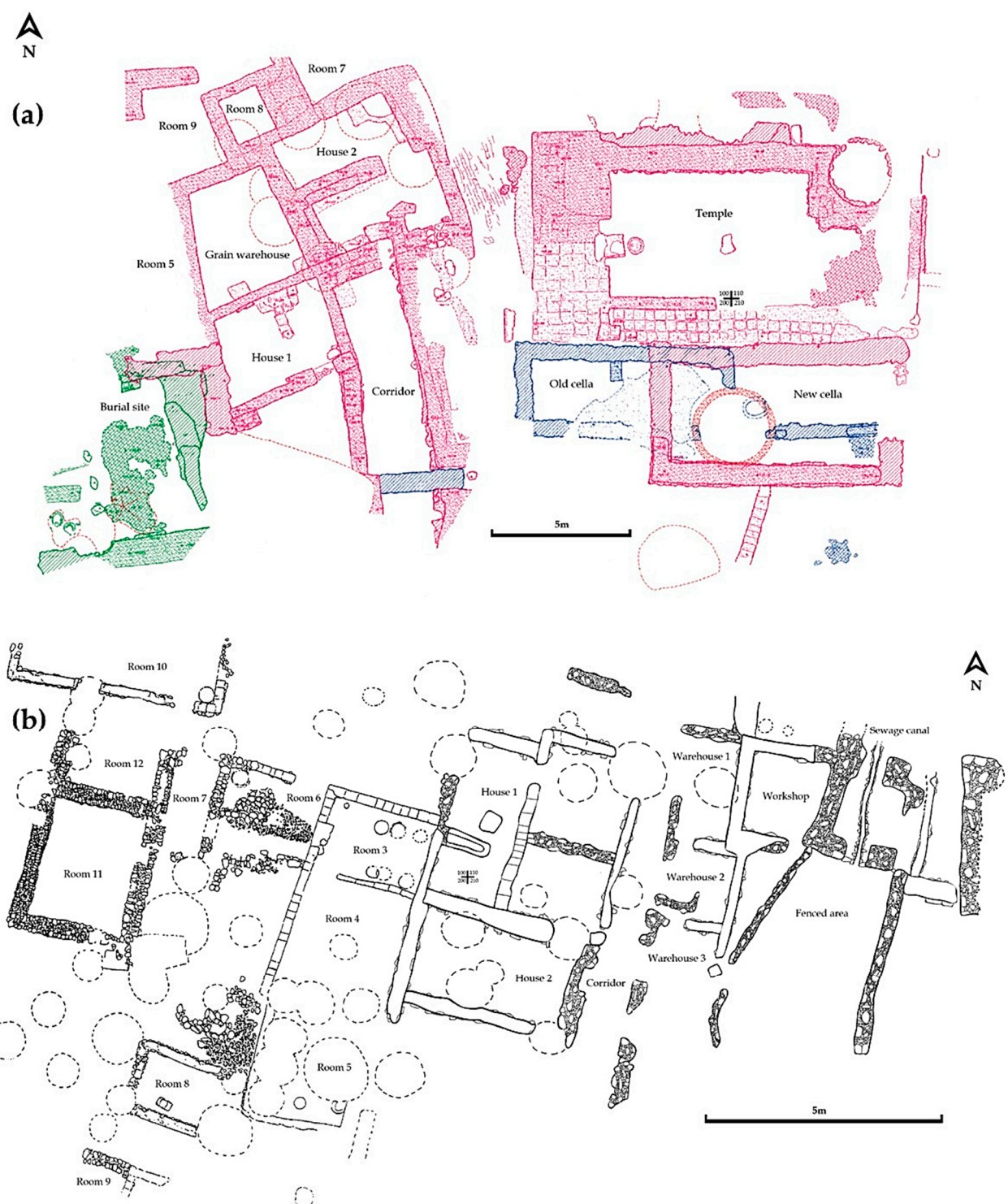

**Figure 2.** General planimetry of the Tell Khamîs excavation: (**a**) **Middle Bronze Age**. In this period, the site was occupied by a temple and a series of annexes. First phase (in blue): remains of a double cella and external offerings table to the south of it; main phase (in pink): a religious building or temple on a terrace (consisting of cella, an interior porticoed courtyard, and a corridor paved with adobe), a series of servants' lodgings of the temple, a corridor or lane separating temple and lodgings, and a large trapezoidal silo. From one of them, House 2, there was direct access to the terrace; in the southwest end, some remains (in green) of Early Bronze III, barely excavated, suggest the existence

of a previous temple. (**b**) **Assyrian.** First occupation of the Assyrian period, two three-room houses (House 1 and 2) in the eastern half of the site and to the east of these, separated by a narrow street, some facilities of agricultural interest on the slope of the Tell: several warehouses and workshops. As the occupation progressed, the western half of the site began to be used for unplanned constructions (Rooms 6 to 12). **References in form of crosses for the excavation grid:** Bronze Age, in the southern area of the temple; Assyrian, in the central part of the image. **Scale** = 5 m.

The partially excavated Early Bronze II (Khamîs XI) only provided three small silos, the corner of a domestic structure, a tomb (reusing one of the silos), and a large stone platform that has been interpreted as a cultural character structure [11] (Figure 2a, green). Middle Bronze II (Khamîs X) (Figure 2a, pink, and Figure 3) corresponds to a time when the entire tell was occupied by a temple and its contiguous rooms (priests' rooms and a large rectangular silo) [13,14]. This level, which consisted of a temple and annexed domestic structures, was looted and destroyed by fire, so that plant macro-remains were abundant and well preserved via carbonization. It is worth noting that many of these were found in large concentrations both in the silo/barn of the temple and in the vessels found in the pantries of the associated buildings. Two [14]C samples from the wooden beams of the dwellings give, with a probability of 95.4% and 94.4%, the calibrated dates of BC 1737–1514 and BC 1688–1445, although according to the ceramic typology, it was proposed that this phase belonged to the Middle Bronze Age I (2000–1850 BC) [12].

Almost 800 years later, the hill was occupied again with a three-nave temple corresponding to the Aramean level (Khamîs IX) that was also destroyed and razed to the ground, perhaps in the context of the campaigns of Shalmaneser III against the Aramean kingdom of Bit Adini [15]. With the next Assyrian level (Khamîs VIII) (Figure 2b), there is a change in the use of the site, since its cultic character disappears to become an agricultural village that was abandoned at the end of the seventh century BC [16] during Persian rule. At first, the Tell will remained unoccupied, except for the existence of two tombs, one of them with a ceramic sarcophagus and two alabaster unguentaries (Khamîs VII) [17]; however, an agricultural village immediately regenerated (Khamîs VI) that persisted for much of the Hellenistic period (Khamîs V to Khamîs II), until the second century BC [18], when the habitat was definitively abandoned until the Islamic period (Khamîs I), when it was used as a cemetery [19].

### 2.2. Recovery and Identification of Plant Remains

Archaeological excavations, carried out between 1992 and 2000, produced 88 sediment samples extending over a chronological range between the Early Bronze Age (third millennium BC) and the Persian dominion of the area (c. fifth century BC). All the archaeobotanical analyses discussed in this contribution share the same guided sampling methodology according to the interpretive potential of the different stratigraphic units (Supplementary Table S1). With the experience of the work in the collection, and the processing and study of the paleobotanical remains of the nearby site of Qara Qûzâq [20,21], it was decided to collect sediment samples from silos and pits, hearths, tombs, and the unbroken vessels. From the former, the volume collected was 20 L, while from the others, the entire sediment was processed.

All carpological remains were recovered via simple flotation through a 0.2 mm mesh sieve during the course of the excavation campaigns. The volume of sediment used for flotation varied from 0.5 L to 12 L. The processed samples were allowed to dry gently in a place close to the excavation and prepared for transport. Most of the material recovered (71% in terms of samples) had been preserved for centuries through carbonization, although a small part had been preserved through mineralization or drying (notably Boraginaceae fruits).

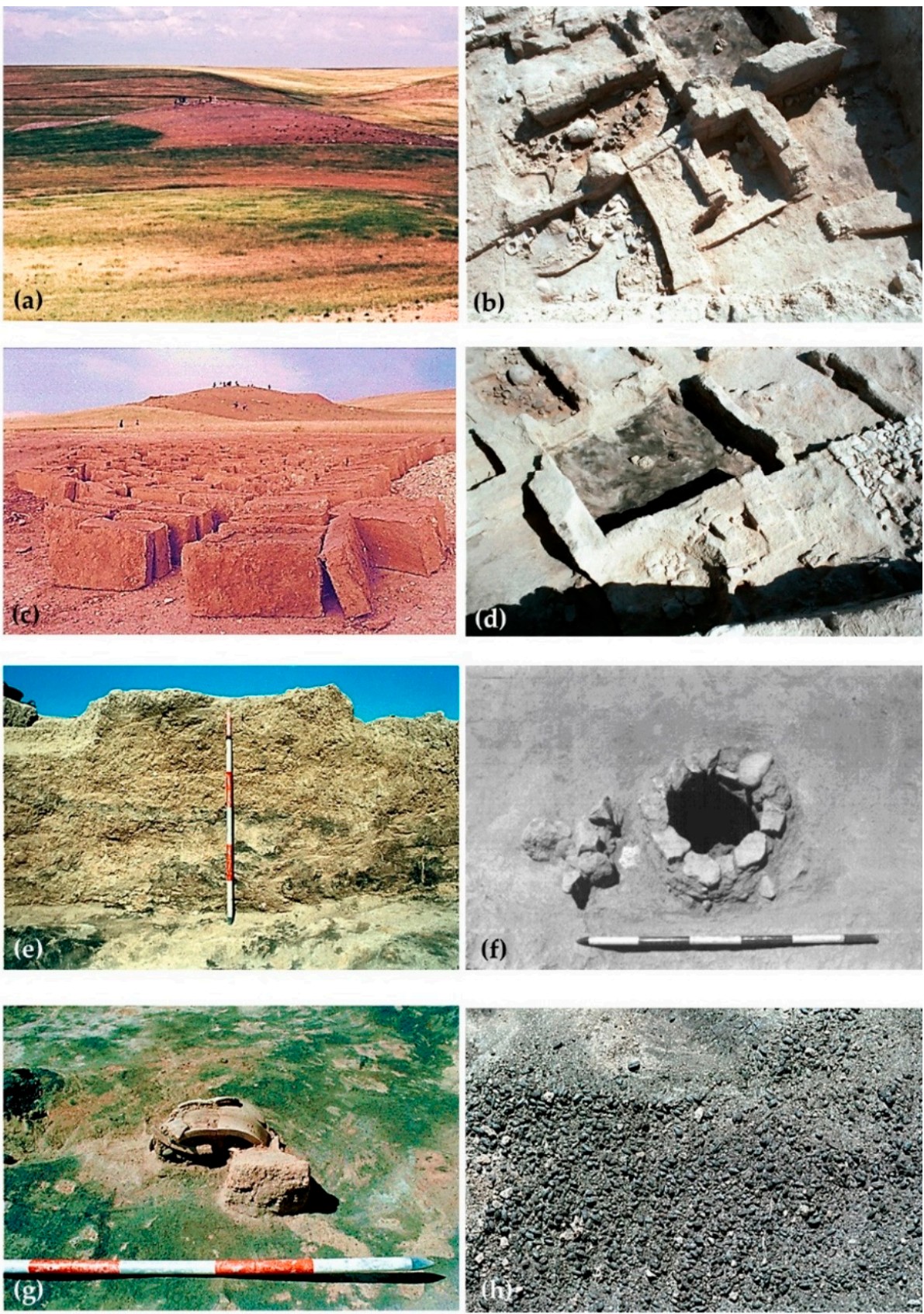

**Figure 3.** Images of Tell Khamîs: (**a**) Tell Khamîs at the beginning of the excavation works (April 1993). The only appreciable difference between the site and the surrounding area is the absence of

vegetation; (**b**) Tell Khamîs, Middle Bronze Age. In the foreground, view of the priests' quarters destroyed by fire and with all the ceramic and plant material inside. In the background, the silo-granary of the temple; (**c**) view of Tell Khamîs (1994) from the small wadi that runs through the valley, from where the small elevation of the hill is more visible. In the foreground, sun-dried mud-bricks made in the same way as those found at the site; (**d**) Tell Khamîs, Middle Bronze Age. In the center, the silo-granary of the temple. In the foreground, the place from which the interior was accessed. Around it, the priests' quarters; (**e**) Tell Khamîs, Middle Bronze. West wall of the temple silo-granary from inside the temple. The place from which the barley was poured coincides with the maximum height of the wall; (**f**) Tell Khamîs, Hellenistic. Mouth of silo 989. This is a silo excavated in the earth and shaped like a bottleneck at the top. Although no plant material was found inside, except for the straw that was used to insulate the grain from the walls, this silo allows us to know what the storage system was; (**g**) Tell Khamîs, Middle Bronze Age. Interior of the silo-granary of the temple with the charred barley and the vessel used to measure the grain extracted; (**h**) Tell Khamîs, Middle Bronze. Interior of the silo-granary of the temple. Detail of the charred plant contents. Images (**a**–**h**): photographs from the Cepoat Archive—Jesús Gómez Carrasco, with permission.

Classification and identification were carried out under a low-power Olympus VMT stereoscopic microscope and a magnification range of 10× to 40× in the Phanerogamy laboratory of the Faculty of Biology of the University of Murcia. The identifications were based on the modern comparative collection of seeds and fruits of the Ethnobotany laboratory, as well as seed atlases and monographs [22–27] and illustrations published in articles presenting the results of archaeobotanical investigations in Syria and neighboring areas [28–33]. Particularly useful for legumes were the study on Ahihud (Israel) by Caracuta et al. [34], Günes and Ali in Turkey [35] and Arranz et al. [36]; on *Pisum* by Kosterin [37,38]; on *Lathyrus* consumption in Late Bronze by Mahler-Slasky and Kislev [39]; on *Lens* seed variability by Punia et al. [40]; on *Astragalus* [41,42]; and, finally, on *Vicia* by Han et al. [43]. For the identification of the remains of the genus *Galium* (Rubiaceae), we took into account the differences in the surface cell patterns between fruits (mericarps) with intact cover and those of seeds and of fruits lacking the outer part of the cover, following Khalik et al. [44] and Jacquat [27].

In the case of *Triticum parvicoccum* Kislev, there are problems with the valid publication of the taxon and the acceptance of the taxon as a separate species, as a subspecies of *Triticum turgidum* L., or simply, as a synonym of the latter species. All these problems are addressed in a joint work of our team with Prof. Kislev from Bar-Ilan University and Dr. Goncharov from the Institute of Cytology and Genetics, Siberian Branch of Russian Academy of Sciences, which is currently in the process of publication. In this present paper, we refer as *T. parvicocum* Kislev to name the very small seeds of about 3 mm in length recovered from Tell Khamîs which would otherwise be called *Triticum aestivum/durum*.

For the correct verification of the nomenclature of the identified species, the POWO [45] database was used. However, for cultivated plant species, we followed, with preference, the taxonomy and nomenclature of ARS-GRIN Taxonomy [46]. Fragments not assigned to any family, species, or genus were classified as indeterminate. The taxa identified and samples from which they were derived, as well as the quantity of remains (charred or dried), and whether they were whole or fragments, are presented in Supplementary Table S2.

The SEM analyses were conducted in the Scientific and Technical Research Area of the University of Murcia. The microstructure of archaeological seeds was investigated by means of field-emission scanning electron microscopy (FE-SEM.) In order to obtain semi-quantitative element results, energy-dispersive x-ray spectroscopy (EDX) was conducted at the locations of interest using field-emission scanning electron microscopy (SEM) with energy-dispersive X-ray spectrometry (EDX). Specimens were mounted on aluminum stubs and platinum sputter-coated with 5.0 nm thin layer (Leica EM ACE 600). Samples were examined using FE-SEM (ApreoS Lovac IML Thermofisher) with a selected voltage of 10 kV and 0.20 nA for imaging and coupled with an Octane Plus EDAX microanalyzer (AMETEK, USA) at 20 kV for EDX analysis.

### 2.3. Analysis of the Habitats Represented

Each plant species or genus has a characteristic pattern of habitats in which it usually grows. This information can be found in the various local floras. In the present study, we preferentially used the Floras of Syria by Post and Dinsmore, and by Mouterde, and the Flora of Israel by Avinoam Danin and Ori Fragman-Sapir [47–49]. Based on the list of taxa identified, we recognized thirteen habitat types. They can be roughly classified into three main types:

- Anthropogenic, with a significant influence of human activity (1. Cultivated areas (crops), 2. cultivated areas (weeds), 3. disturbed habitats, and 4. nutrient-rich soils, ruderal).
- Natural or semi-natural (5. batha, 6. phrygana, 7. shrub-steppes, 8. sand, 9. hard rock outcrops, and 10. mountain steppe forest)
- Specialized (11. salty habitats, 12. desert, 13. humid habitats).

To the aforesaid must be added the "Imported" category, those very remote habitats unlikely present in the area, characteristic of the Mediterranean coast, the tops of high mountains or tropical habitats. This analysis will make it possible to determine the predominant flora profiles and, based on this, the characteristics of the environment and possible changes over time in the area of the site. Furthermore, by applying the same analysis to different sites, depending on the flora shared with Tell Khamîs, we can detect common and differential patterns, especially in relevant aspects such as the importance of anthropized habitats or salinization. The habitats of the different taxa identified are listed in Supplementary Table S3.

The relevance of habitats can be represented over the periods analyzed in terms of the plant remains identified from Tell Khamîs. The diachronic evolution can be calculated as: (a) the percentage of genera and species growing in each habitat type out of the total number of genera and species identified in each period; (b) the contribution of each period to the total number of species inhabiting each habitat type, expressed as a percentage; 0 means no occurrence in that period, and 100 means that all species of a determinate habitat occurred in that period.

### 2.4. Comparison with Other Sites Using Multivariate Analysis Techniques

2.4.1. Comparison Based on Presence/Absence of Species

If we want to obtain an idea of the position of Tell Khamîs within the macro-regional context or, at least, that of the Euphrates valley and its evolution, it is necessary to compare the data obtained (the list of species) with those of other sites. We also aim to determine whether there are culturally characteristic subgroups within the cultivated species. Once published by the various authors and considering the levels of reliability of the identifications at species or genus level, it is clear that an analytical tool such as multivariate analysis can simplify the process of comparison that all the authors carry out when they try to present a synthesis or an overall view. There are certainly aspects to be considered for the botanical identification of plant remains; these include the availability of an extensive reference collection, the previous experience of the team in the identification of remains, or the knowledge of the regional flora. Wild species tend to be identified following the principle of parsimony or Ockham's razor, based on currently available local floras, generic seed atlases, and previously published archaeobotanical literature; however, this is also carried out when identifying cultivated species. It is, therefore, to be expected that when the sites studied are located in biogeographically distant areas, they only share with Tell Khamîs the repertoire of cultivated species, or a specific part of it; meanwhile, those geographically closer present a greater number of shared species that also include a substantial number of wild species.

The comparison with reference sites (Supplementary Table S3) is made on the base of the species identified in Tell Khamîs by means of a presence–absence matrix, whereby each row represents a species, each column represents a site, and the elements of the matrix are $d(i,j) = 1$ if species i is present at the site j, and $d(i,j) = 0$ otherwise. The sum of elements

along a row yields the number of sites in which the corresponding species occurs (that is, its range size $n_i$), and the equivalent sum along a column equals the total number of species present at a site (that is, its shared species richness $s_j$). The fill of the matrix is the total number of species occurrences. The diversity field volume ($D_i$) of species i is the summation of species-richness values for sites ($s_j$) within its range [50,51]. The range–diversity values ($s_i$) for each species is expressed as the ratio $D_i/n_i$. The dispersion field volume ($R_j$) of site j is the summation of range sizes of the species occurring at that site. The per-site range size value $n_j$ is expressed as the ratio $R_j/s_j$). The sites selected are those with at least fourteen species shared with Tell Khamîs (mean 18, max 59) among those included in the exhaustive review by Rivera et al. [52].

The crude matrix of the presence/absence of taxa by site was used to compute an intersite dissimilarity matrix using Darwin 6 V.6.0.21 (26 April 2019) [53,54]. The Sokal–Sneath dissimilarity index was calculated (un2) using Equation (1).

$$d_{ij} = 2(b + c)/a + 2(b + c) \tag{1}$$

where $d_{ij}$ is the dissimilarity between samples i and j, a: the number of variables where $x_i$ = presence and $x_j$ = presence; b: the number of variables where $x_i$ = presence and $x_j$ = absence; and c: the number of variables where $x_i$ = absence and $x_j$ = absence. Dissimilarities are even and are Euclidean distances. The dissimilarity is =0 for two samples sharing the 202 species and =1 for two samples which present 0 species shared. This index concerns 'presence/absence' data where only the 'presence' modality is informative, with the modality 'absence' mainly expressing an absence of information. These two modalities are not symmetrical and their exchange leads to a completely different dissimilarity value. This index considers that a common absence for two units is uninformative to measure their dissimilarity [54,55]. Therefore, similarity here reflects the number of coinciding species, and dissimilarity is inversely proportional to this.

These pairwise dissimilarities can be represented in a multidimensional space, but in order to obtain meaningful graphic representation of these relationships in a two-dimensional plane, we used cluster analysis [55].

We used the agglomerative hierarchical method that arranges the clusters into a hierarchy so that the relationships between different groups are apparent. Minimum variance clustering (Ward's method) focuses on determining how much variation is within each cluster. In this way, the clusters will tend to be as distinct as possible, since the criterion for clustering is to have the least variation [55].

Principal coordinate analysis (PCoA) is a member of the factorial analysis family working on distance matrices. It considers the space of high dimension defined by the distances between units two-by-two. This space has too high a dimension to be readable so PCoA searches for a subspace of low dimension where distances between units are as close as possible to the original distances. Working with the above dissimilarity matrix, we represent the units against the first and second axes. Details of the PCoA procedure are described at [53].

As the preliminary dissimilarity analysis resulted in non-Euclidean distances, given the extraordinary diversity of occurrence columns (species shared with Tell Khamîs). This led us to reduce the comparison in a second analysis with a smaller range of sites, with sj values equal to or greater than 15. As a consequence, the number of sites was reduced from 77 to 31.

### 2.4.2. Comparison Based on Relevance of the Different Habitat Types Based on the List of Species

In addition to the comparative analysis based on the presence–absence matrix of species, the representation of the different habitat types in the list from each site was calculated in terms of the percentage of shared taxa that usually occurs in each one of the habitat types. This approach results in a normalized-type matrix. Thus, relevant archaeological sites of Syria, Turkey, Iraq, and Iran that present the species identified in

Tell Khamîs were analyzed, and the values of the main habitats represented by the list of species were calculated in terms of normalized proportion with respect to the maximum. Among the site parameters, $s_{sj}$ represents the shared species-richness values for sites; $R_j$: the dispersion field volume of a site j is the summation of range sizes of the species occurring at that site; and $n_j$ is the per-site range size value expressed as the ratio $R_j/s_j$).

We expected to differentiate the groups of sites and determine their relationships with respect to Tell Khamîs as a function of the more represented habitats using multivariate analysis. For the purpose of comparing and classifying the different sites, the chi-square dissimilarity index was calculated (which is optimal for such a type of data). This measure expresses a value $x_{ik}$ as its contribution to the sum $x_i$ for all variables and is a comparison of unit profiles [53–55]. Further, a hierarchical tree was constructed using the Ward's minimum variance criterion [55].

## 3. Results

### 3.1. Taxa Identified

3.1.1. General Results

A total of 20,606 whole remains and 37,646 seed and fruit fragments belonging to 92 taxa and 35 different plant families were identified (Figures 4–8 and Tables 1 and 2). The Middle Bronze Age and Assyrian levels are the richest in taxa in general and also in exclusive or nearly exclusive taxa (50% or more of the finds in that period) (Table 1). In terms of total remains and also in terms of number of remains per taxon, these are also the most important, with the Middle Bronze Age levels standing out (Table 1).

**Table 1.** Numbers of fruit and seed remains from Tell Khamîs.

| Period\Numbers | Entire Seeds and Fruits | | Seeds and Fruits (Excl. *Hordeum*) * | Remains/Taxon | Remains/Taxon (Excl. *Hordeum*) * | Fragments | | Fragments (Excl. *Hordeum*) * | Exclusive Taxa (≥50% Findings in This Period) |
|---|---|---|---|---|---|---|---|---|---|
| | Total | Taxa | Total | Average | Average | Total | Taxa | Total | % |
| Early Bronze | 45 | 11 | 27 | 4.7 | 2.2 | 14 | 2 | 2 | 27.3 |
| Middle Bronze | 17,435 | 58 | 1797 | 496.1 | 19.9 | 36,493 | 6 | 808 | 50 |
| Aramaic | 15 | 5 | 15 | 3 | 3 | 0 | 0 | 0 | 20 |
| Assyrian | 1899 | 58 | 1568 | 11.6 | 8.5 | 905 | 8 | 53 | 42.6 |
| Assyrian– Hellenistic– Persian, | 705 | 49 | 610 | 4.3 | 3.6 | 139 | 7 | 16 | 42 |
| Hellenistic– Persian | 507 | 38 | 264 | 7.2 | 3.1 | 95 | 3 | 10 | 26.3 |
| Totals | 20,606 | 92 | 4281 | 109.4 | 8.1 | 37,646 | 16 | 889 | - |

\* Notice: excl. *Hordeum* = numbers excluding cultivated barley of the genus *Hordeum*.

Species do not occur uniformly throughout the various periods and in the various samples. A substantial number of taxa (eighty-seven) occur in fewer than five samples, while only three (*Hordeum vulgare* L.; *Buglossoides tenuiflora* (L.f.) I.M.Johnst.; and *Buglossoides arvensis* (L.) I.M.Johnst., Figures 5 and 6) occur in more than twenty samples. The most relevant plant families in terms of number of taxa or number of remains are Poaceae, Boraginaceae, Leguminosae, and Caryophyllaceae (Table 2).

The number of whole remains recovered from each taxon is generally relatively small, with 55 of the 92 identified occurring in fewer than 5. Some are better represented, with more than 100 seeds or fruits recovered from eight taxa. The 88 samples from contexts stratigraphically comprised between the Early Bronze Age and the Persian period presented numerous carpological remains in an excellent state of preservation belonging to over ninety different plant taxa and 35 different families (Table 2). However, in some cases, identification beyond the genus level was not possible.

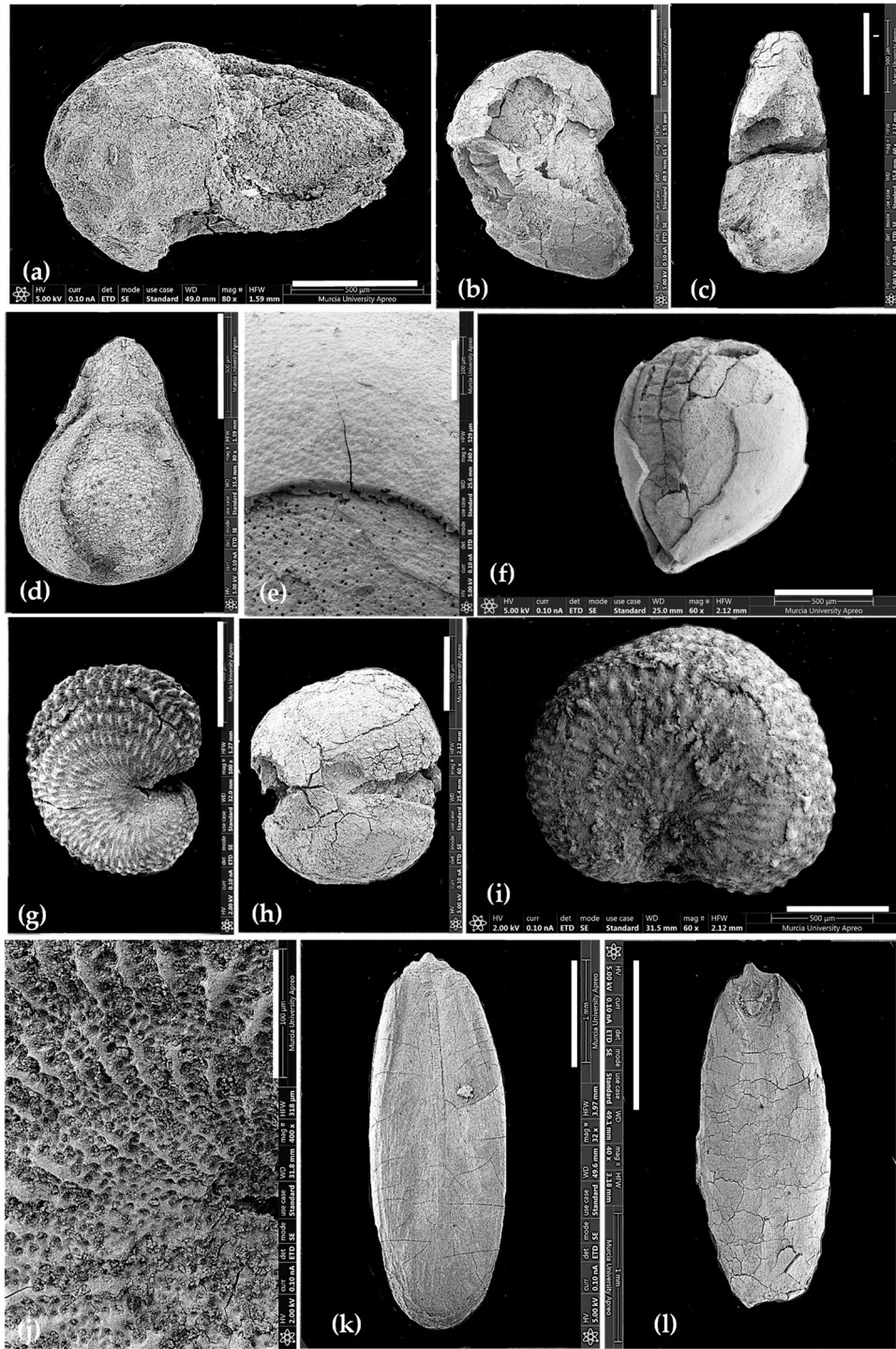

**Figure 4.** Seed remains from Tell Khamîs. LAMIACEAE: (**a**) *Ajuga* sp., 040 LP; (**b**) *Ajuga* sp., 010 HP. VALERIANACEAE: (**c**) *Valerianella coronata* (L.) DC. (deteriorated) 029 LP; (**d**) *Valerianella vesicaria* Moench, 001 LP. CYPERACEAE: (**e**) *Bolboschoenus maritimus* (L.) Palla (surface), 103 MB; (**f**) *Bolboschoenus maritimus* (L.) Palla, 103 MB. CARYOPHYLLACEAE: (**g**) *Silene* sp. (incl. *Silene vulgaris* (Moench) Garcke), 004 LP; (**h**) *Gypsophila vaccaria* (L.) Sm. (=*Vaccaria pyramidata* Medik.), 076 MB; (**i**) *Silene* sp. (incl. *Silene vivianii* Steud.), 038 LP; (**j**) *Silene* sp. (incl. *Silene vulgaris* (Moench) Garcke) (surface), 004 HP. POACEAE: (**k**) *Lolium* sp. (large, ventral view), 027 LP; (**l**) *Lolium* sp.(small, dorsal view), 027 LP. Abbreviations: AS—Assyrian, EB—Early Bronze, HP—Hellenistic–Persian, LP—Later Period, MB—Middle Bronze. Scale Bar: (**k**,**l**) 2 mm; (**a**–**d**,**f**–**i**) 500 µm; (**e**,**j**) 100 µm.

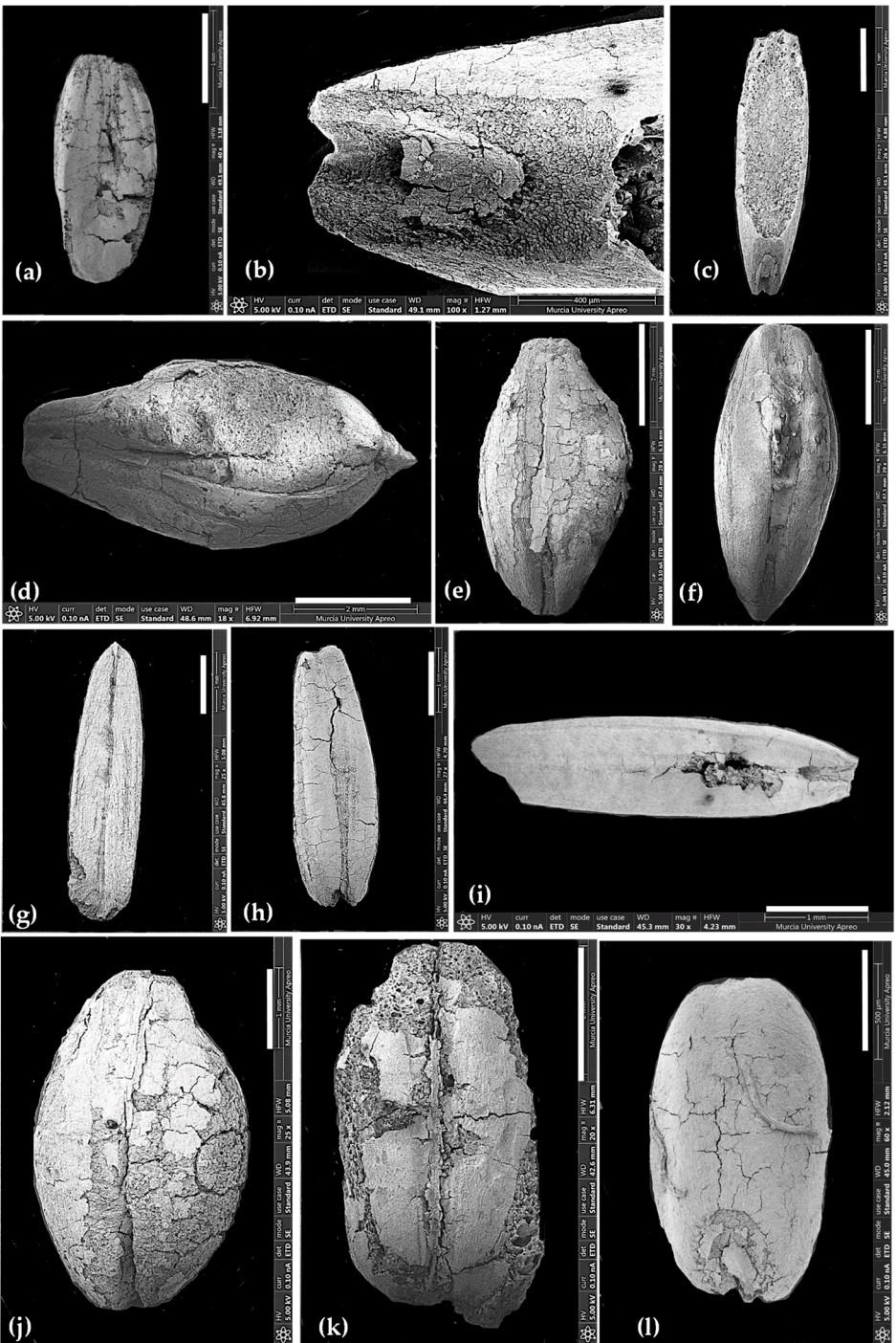

**Figure 5.** Seed remains from Tell Khamîs. POACEAE: (**a**) *Lolium* sp., 096 AS; (**b**) *Bromus danthoniae* Trin. ex C.A.Mey. (detail), 079 EB; (**c**) *Bromus danthoniae* Trin. ex C.A.Mey., 079 EB; (**d**,**e**) *Hordeum vulgare* L. var. *vulgare*, 061 MB; (**f**) *Hordeum vulgare* L. var. *vulgare*, 014 HP; (**g**) *Bromus sterilis* L., 072 MB; (**h**) *Hordeum spontaneum* K.Koch, 075 MB; (**i**) *Taeniatherum* sp., 076 MB; (**j**) *Hordeum distichon* L. var. *nudum* L. (= *Hordeum vulgare* var. *nudum* Spenn.), 092 AS; (**k**) *Triticum aestivum* L./*T. turgidum* L. (naked hexaploid or tetraploid wheat), 085 AS; (**l**) *Triticum* cf. *parvicoccum* Kislev., 003 AS. Abbreviations: AS— Assyrian, HP—Hellenistic–Persian, LP—Later Period, MB—Middle Bronze. Scale Bar: (**d**–**f**,**k**) 2 mm; (**a**,**c**,**g**–**j**) 1 mm; (**l**) 500 μm; (**b**) 400 μm.

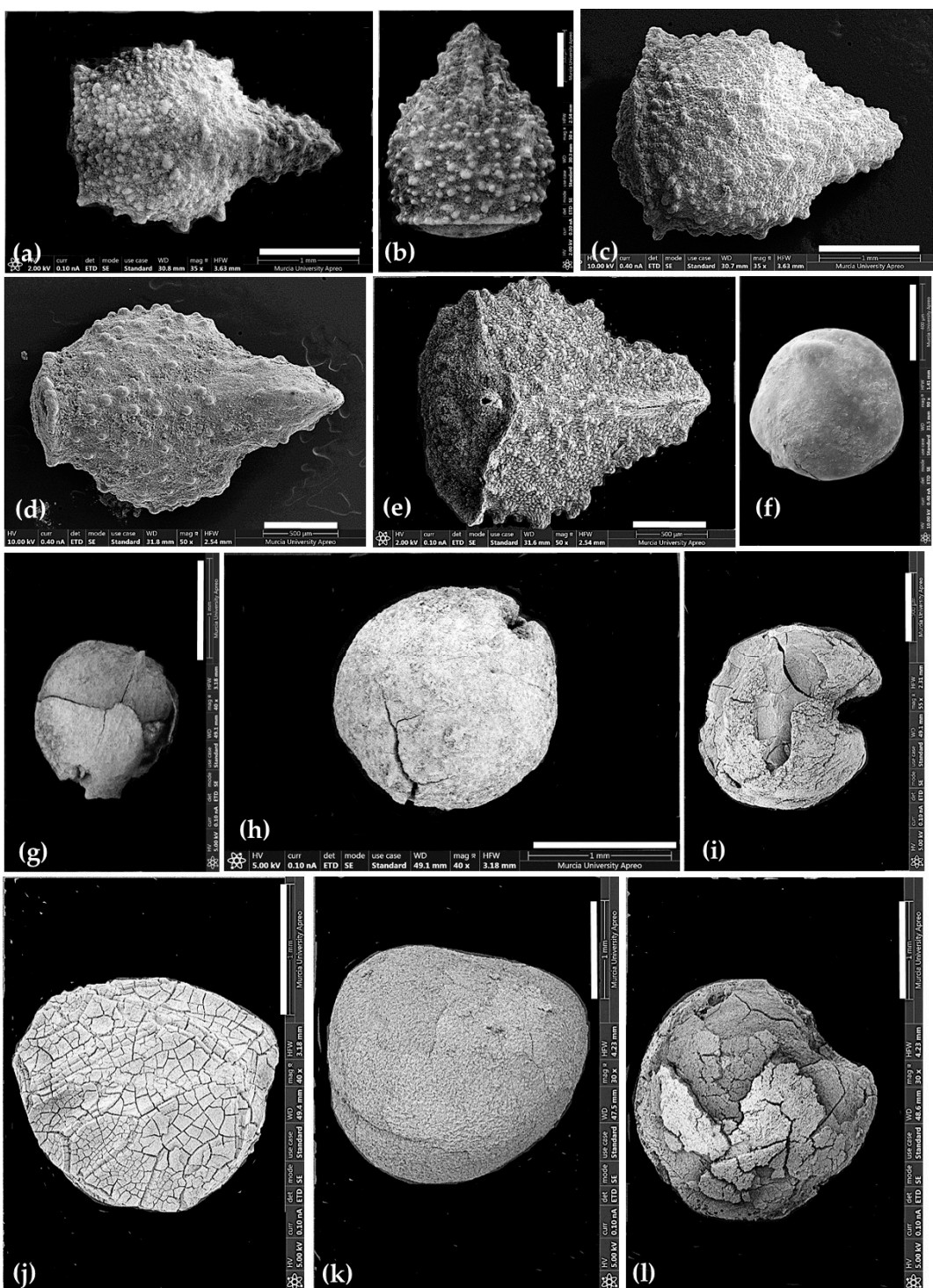

**Figure 6.** Seed remains from Tell Khamîs. BORAGINACEAE: (**a**) *Arnebia decumbens* Coss. & Kralik, 015 HP; (**b**) *Buglossoides arvensis* (L.) I.M.Johnst., 004 HP; (**c**) *Arnebia linearifolia* DC., 010 HP; (**d**) *Buglossoides tenuiflora* (L.f.) I.M.Johnst., 004 HP; (**e**) *Buglossoides arvensis* (L.) I.M.Johnst., 071 MB; (**f**) AMARANTHACEAE *Amaranthus* sp., 005 LP. PAPAVERACEAE: (**g**) *Fumaria* sp., 075 MB; (**h**) *Fumaria* sp., 010 HP. MALVACEAE: (**i**) *Malva* sp., 091 AS. FABACEAE: (**j**) *Lens culinaris* subsp. *orientalis* (Boiss.) Ponert, with heavily cracked surface as a result of degradation, 351 LP; (**k**), *Vicia ervilia* (L.) Willd., 001 LP; (**l**) *Vicia ervilia* (L.) Willd., 076 MB. Abbreviations: AS—Assyrian, EB—Early Bronze, HP—Hellenistic–Persian, LP—Later Period, MB—Middle Bronze. Scale Bar: (**a**,**c**,**f**–**h**,**j**–**l**) 1 mm; (**b**,**d**,**e**,**i**) 500 μm.

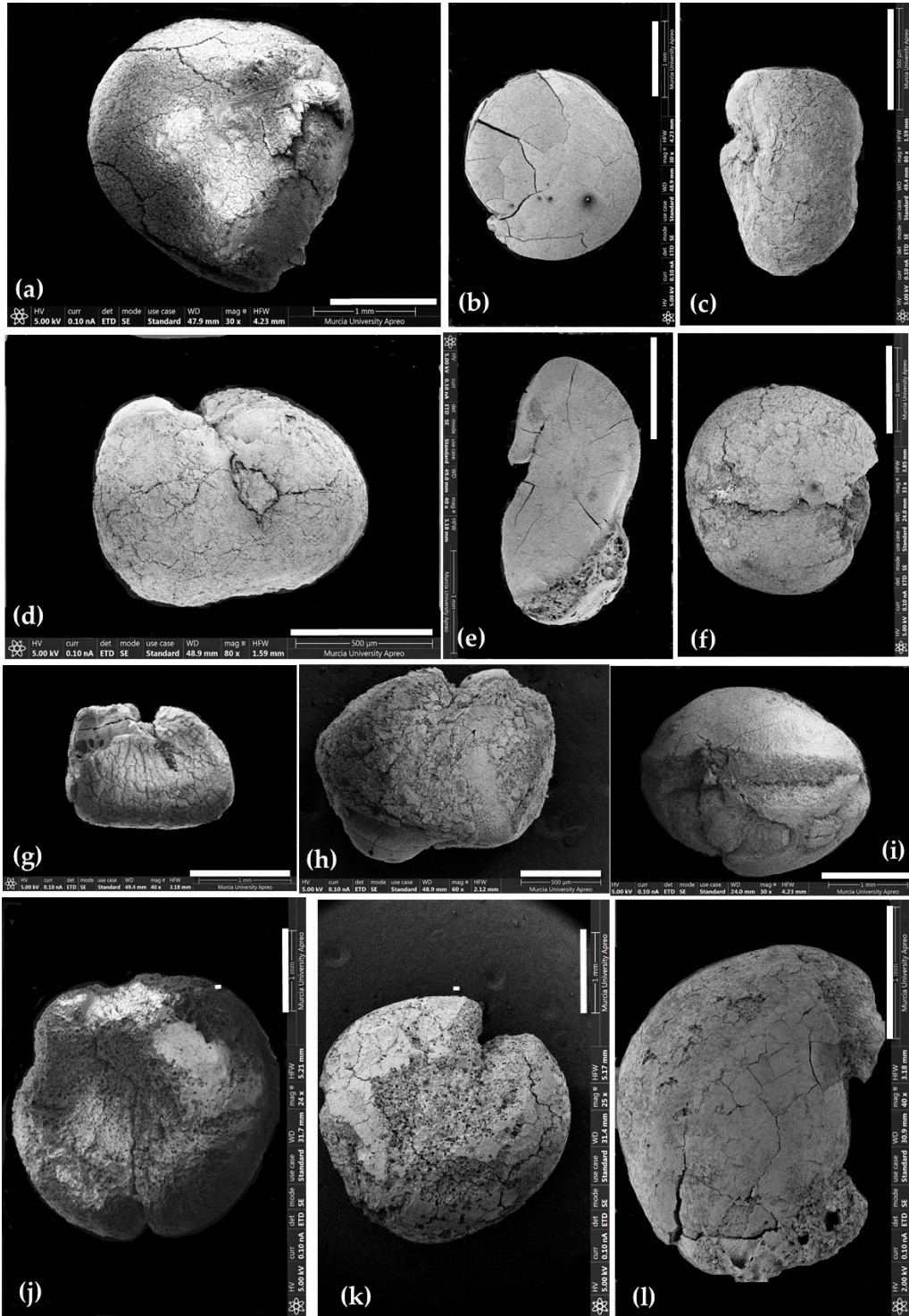

**Figure 7.** Seed remains from Tell Khamîs. FABACEAE: (**a**) *Vicia ervilia* (L.) Willd., 076 MB; (**b**) *Lens culinaris* Medik. subsp. *orientalis* (Boiss.) Ponert, with well-preserved surface, 015 HP; (**c**) *Trigonella* cf. *coerulescens* (M.Bieb.) Halácsy, 001 LP; (**d**) *Trigonella* cf. *coerulescens* (M.Bieb.) Halácsy, 004 HP; (**e**) *Medicago* sp., 001 LP; (**f**) *Pisum* sp. (seed), 073 MB; (**g**) *Trigonella* cf. *coerulescens* (M.Bieb.) Halácsy, 001 LP; (**h**) *Astragalus* sp. (rhombiform type), 089 AS; (**i**) *Vicia palaestina* Boiss., 073 MB; (**j**) *Lathyrus cicera* L., 076 MB; (**k**) *Lens* cf. *culinaris* Medik., 015 HP; CAPPARACEAE (**l**) *Capparis zoharyi* Inocencio, D.Rivera, Obón & Alcaraz, 082 AS. Abbreviations: AS—Assyrian, HP— Hellenistic–Persian, LP—Later Period, MB—Middle Bronze. Scale Bar: (**a**,**b**,**e**–**g**,**j**–**l**) 1 mm; (**c**,**d**,**h**) 500 µm; (**i**) 300 µm.

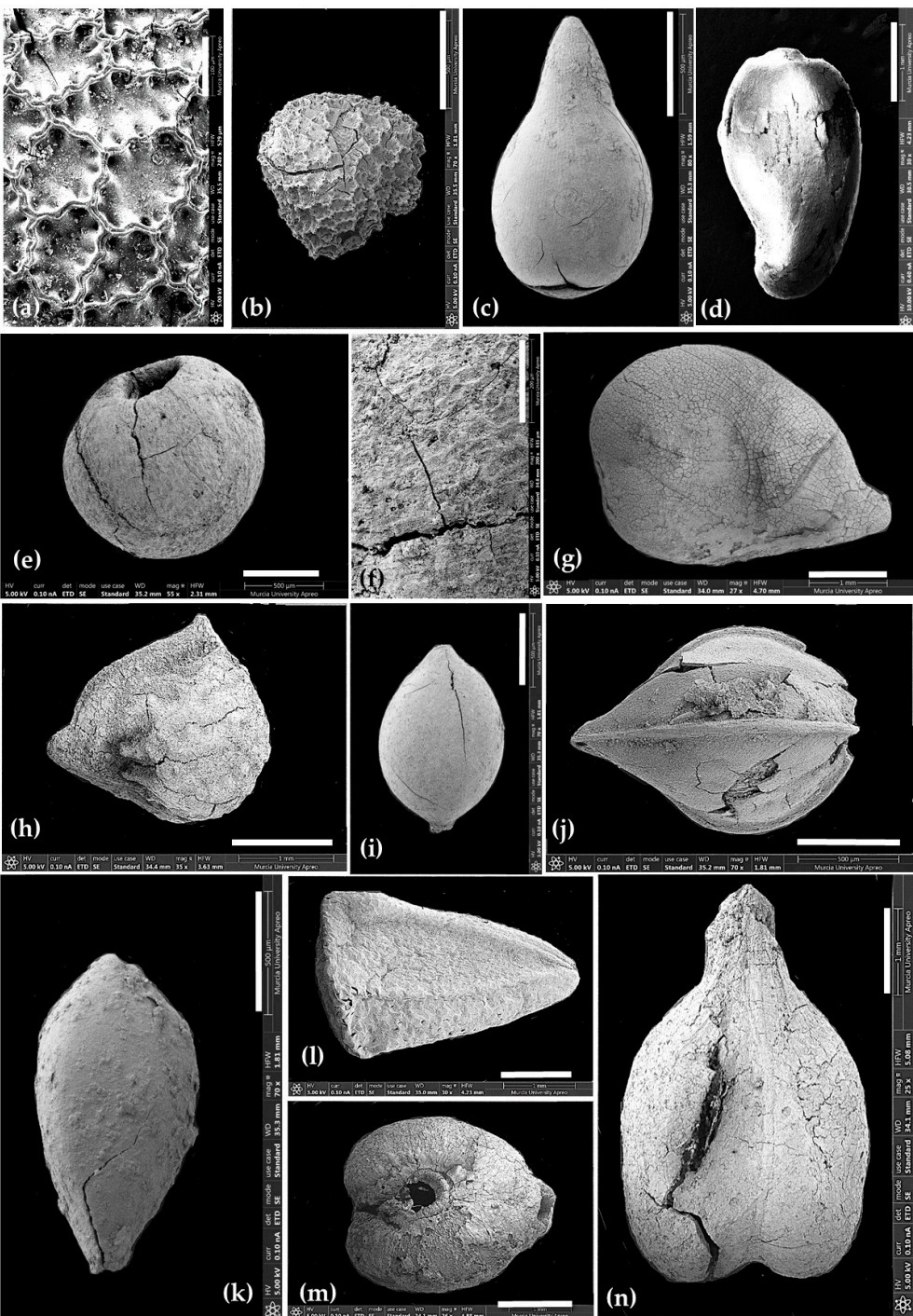

**Figure 8.** Seed remains from Tell Khamîs. SOLANACEAE: (**a**) *Hyoscyamus aureus* L., 027 LP; (**b**) *Hyoscyamus aureus* L., 027 LP. THYMELAEACEAE: (**c**) *Thymelaea passerina* (L.) Coss. & Germ., 004 LP; ASTERACEAE; (**d**) *Centaurea iberica* Trevis. ex Spreng., 068 MB. RUBIACEAE: (**e**) *Galium aparine* L. s.l. (seed), 073 MB; (**f**) *Galium aparine* L. s.l. (seed surface detail), 073 MB. ROSACEAE: (**g**) *Pyrus syriaca* Boiss., 073 MB. RANUNCULACEAE: (**h**) *Adonis aleppica* Boiss., 010 HP; POLYGO-NACEAE (**i**) *Polygonum arenarium* Waldst. & Kit. (=*Polygonum venantianum* Clem.), 001 LP; (**j**), *Rumex* cf. *crispus* L., 073 MB; (**k**), *Polygonum arenarium* Waldst. & Kit. (=*Polygonum venantianum* Clem.), 005 LP. NITRARIACEAE: (**l**) *Peganum harmala* L., 087 AS; VITACEAE; (**m**) *Vitis vinifera* L., 085 AS; (**n**) *Vitis vinifera* L., 108 AS. Abbreviations: AS—Assyrian, HP— Hellenistic–Persian, LP—Later Period, MB—Middle Bronze. Scale Bar: (**d,g,h,l–n**) 1 mm; (**b,c,e,i–k**) 500 μm; (**f**) 200 μm; (**a**) 100 μm.

**Table 2.** Most relevant plant families, in numbers of remains, represented in Tell Khamîs.

| Families [1] | Taxa | Remains (Only Entire, Excluding Fragments) |
|---|---|---|
| Poaceae | 21 | 16,747 |
| Boraginaceae | 5 | 2397 |
| Leguminosae | 15 | 839 |
| Caryophyllaceae | 6 | 91 |
| Nitrariaceae | 1 | 86 |
| Polygonaceae | 4 | 68 |
| Rubiaceae | 2 | 65 |
| Pedaliaceae | 1 | 51 |
| Amaranthaceae | 3 | 45 |
| Capparaceae | 1 | 26 |
| Vitaceae | 1 | 22 |
| Valerianaceae | 2 | 21 |
| Cyperaceae | 3 | 19 |
| Fabaceae | 2 | 19 |
| Ranunculaceae | 1 | 15 |
| Brassicaceae | 5 | 13 |
| Malvaceae | 2 | 12 |
| Asteraceae | 2 | 11 |
| Papaveraceae | 2 | 11 |
| Asparagaceae | 1 | 8 |
| Lamiaceae | 2 | 7 |
| Thymelaeaceae | 1 | 7 |
| Euphorbiaceae | 2 | 5 |

[1] Note: families represented in the table are those with 5 or more entire remains.

### 3.1.2. Common Versus Rare Taxa

Twenty or more remains were recovered from only 21 of the 92 taxa identified at Tell Khamîs (Table 3). Of these, barley is particularly noteworthy for the number of remains. This species was found in in 49 different samples, but notably in a silo of about 4 m$^3$ in volume. Considering that the weight of a cubic meter of barley ranges between 600 and 700 kg [56,57], the total weight of barley stored in that silo would exceed 2600 kg.

More than 50,000 barley remains have been recovered (Table 3 and Table S1), of which more than 16,000 are whole and the rest are fragments. If we take into account that the weight of a barley grain ranges between 48 and 52 mg, depending on the varieties [26], the 2600 kg of barley in the silo would be about 52 million grains, so the sample recovered is only one thousandth of the total at the site. The marked disproportion between the numbers of barley remains recovered, and those of the various wheat species leads us to believe that wheat was a rarity in the various levels of Tell Khamîs. This is possibly linked to the greater sensitivity to soil salinity on the part of the various wheats compared to barley, which is much more tolerant and grows in soils with high conductivity [2].

In addition, 27 taxa, wild plants, and crops (*Linum usitatissimum* L., *Morus alba* L., *Sorghum bicolor* (L.) Moench, *Triticum* cf. *parvicoccum* Kislev, and *T. turgidum* subsp. *dicoccum* (Schrank ex Schübl.) Thell.), weeds (*Ammi majus* L., *Ornithogalum* sp., *Bellevalia* sp., *Medicago* sp., *Sinapis* sp., *Glaucium aleppicum* Boiss. & Hausskn. ex Boiss., *Veronica* cf. *syriaca* Roem. & Schult., *Echinochloa* sp., *Panicum miliaceum* L., *Phalaris* sp., *Poa* sp., *Sherardia arvensis* L., *Scrophularia* sp., *Polygonum aviculare* L., and *Reseda luteola* L.), and others (*Alyssum desertorum* Stapf, *Juniperus* cf. *excelsa* M.Bieb., *Cyperus* sp., *Teucrium* sp., *Hordeum distichon* L. var. *nudum* L., *Pyrus syriaca* Boiss., and *Viola* cf. *pentadactyla* Fenzl), are represented by only one seed or fruit, many of which are of considerable interest. Unfortunately, the low representation of these taxa does not allow conclusions to be drawn as to the significance of their presence.

**Table 3.** Most relevant species represented in Tell Khamîs.

| Taxa [1] | Families | SE | SF | Total Remains | Entire | Fragments |
|---|---|---|---|---|---|---|
| *Hordeum vulgare* L. var. *vulgare* | Poaceae | 49 | 48 | 53,082 | 16,325 | 36,757 |
| *Buglossoides tenuiflora* (L.f.) I.M.Johnst. | Boraginaceae | 5 | 4 | 1169 | 1169 | 0 |
| *Vicia ervilia* (L.) Willd. | Fabaceae | 67 | 0 | 1591 | 789 | 802 |
| *Buglossoides arvensis* (L.) I.M.Johnst. | Boraginaceae | 51 | 0 | 545 | 545 | 0 |
| *Heliotropium* sp. | Boraginaceae | 1 | 0 | 432 | 432 | 0 |
| *Lolium* sp. | Poaceae | 7 | 2 | 189 | 165 | 24 |
| *Arnebia linearifolia* DC. | Boraginaceae | 25 | 0 | 148 | 148 | 0 |
| *Arnebia decumbens* Coss. & Kralik | Boraginaceae | 4 | 1 | 105 | 105 | 0 |
| *Taeniatherum* sp. | Poaceae | 20 | 0 | 112 | 97 | 15 |
| *Peganum harmala* L. | Nitrariaceae | 1 | 0 | 86 | 86 | 0 |
| *Silene* sp. | Caryophyllaceae | 22 | 0 | 72 | 72 | 0 |
| *Hordeum spontaneum* K.Koch | Poaceae | 9 | 2 | 72 | 66 | 6 |
| *Galium aparine* L. s.l. | Rubiaceae | 21 | 0 | 65 | 65 | 0 |
| *Sesamum indicum* L. | Pedaliaceae | 8 | 1 | 52 | 51 | 1 |
| *Rumex crispus* L. | Polygonaceae | 13 | 8 | 37 | 37 | 0 |
| *Polygonum arenarium* Waldst. & Kit. (=*P. venantianum* Clem.) | Polygonaceae | 10 | 0 | 31 | 31 | 0 |
| *Eremopyrum bonaepartis* (Spreng.) Nevski | Poaceae | 9 | 0 | 28 | 28 | 0 |
| *Capparis zoharyi* Inocencio, D.Rivera, Obón & Alcaraz | Capparaceae | 8 | 0 | 26 | 26 | 0 |
| *Chenopodium album* L. | Amaranthaceae | 6 | 0 | 25 | 25 | 0 |
| *Vitis vinifera* L. | Vitaceae | 6 | 0 | 44 | 22 | 22 |
| *Adonis aleppica* Boiss. | Ranunculaceae | 9 | 1 | 21 | 19 | 2 |

[1] Note: species represented in the table are those with 20 or more remains. Abbreviations: SE—Samples with entire remains of the species; SF—Samples with fragments of the species.

### 3.2. Species Abundance in the Different Periods

A series of questions about the plant remains must be highlighted in relation to the phases of the site and the nature of the structures in which they were found. Throughout the following paragraphs, we will refer to the species and other taxa that appear as exclusive (100% of the remains appeared in that period) or relevant (50% or more of the remains were identified in the samples of that period).

#### 3.2.1. Bronze Age

The total of samples from the Bronze Age comprises 2 samples for the Early Bronze Age that are derived exclusively from funerary contexts and 17 samples for the Middle Bronze Age that correspond mainly to strata inside residential structures. The Bronze Age is the richest period in terms of unique and relevant species in the Tell Khamîs repertoire. The Early Bronze Age levels yielded, as exclusive or relevant taxa for this level, *Polygonum aviculare* L. (100%) and *Gypsophila* sp. (75%) (Figure 4) of the eleven identified from this period. Only part of a dwelling and a religious structure were excavated from this period to which some children's tombs could be related. In connection with the dwelling, there were three domestic silos, although only silo 275 provided plant material, notably *Hordeum distichon*, *Phragmites australis* (1 reed-stem), and *Vicia ervilia* (Figure 6) [12].

In Early Bronze II, the few silos appear to be domestic in nature, and in their last phase, they were used as dumps; thus, the material found responds mainly to household waste. The tomb that took advantage of one of the silos did not do so in its entirety since it was not completely emptied, leaving remains of the old dump under the corpse; in addition, its surface was reduced by half with the construction of a separation wall.

There is no evidence of pitch contemporary to the Middle Bronze except for a small dump related to the primitive altar of the temple [12,14]. In any case, the fire strata at the time of the destruction contained quite a few plant remains. Something of interest at this level is a silo built of adobe, with a quadrangular floor plan and a surface area of 14 $m^2$ that conserved an elevation close to 2 m, so its minimum capacity was 28 $m^3$. Inside, there

was a little more than 4 m$^3$ of charred *Hordeum distichon* L. (Figure 5) along with a vessel that must have been the measure used to count rations [14]. On the other hand, the rooms of the priests contained all the ceramic trousseaus with their vegetable contents inside [13].

The Middle Bronze Age yielded, as exclusive or relevant, 30 of the 58 taxa identified from this period. Most are segetal: *Ammi majus* L. (100%), *Silene* sp. (100%), *Ornithogalum* sp. (100%), *Bellevalia* sp. (100 %), *Glaucium aleppicum* Boiss. & Hausskn. ex Boiss. (100%), *Phalaris* sp. (100%), *Setaria* sp. (100%), *Poa* sp. (100%), *Scrophularia* sp. (100%), *Taeniatherum* sp. (93%), *Bromus sterilis* L. (90%), *Trigonella caelesyriaca* Boiss. (83%), *Asparagus horridus* L.f. (75%), *Galium aparine* L. s.l. (75%), *Bupleurum subovatum* Link ex Spreng. (66%), *Eremopyrum bonaepartis* (Spreng.) Nevski (60%), *Buglossoides tenuiflora* (L.f.) I.M.Johnst. (51%), *Bromus danthoniae* Trin. ex C.A.Mey. (50%), *Iris aucheri* (Baker) Sealy (50%), *Melilotus* sp. (50%), *Fumaria* sp. (50%), and *Panicum miliaceum* L. (50%). Others are crops or crop relatives: *Hordeum distichon* L. var. *nudum* L. (100%), *Hordeum vulgare* L. var. *vulgare* (95% of the 16325 entire seeds identified), *Vicia ervilia* (L.) Willd. (100% of the 788 entire seeds), *Lens culinaris* Medik. subsp. *microsperma* (Baumg.) N.F. Mattos (66%), *Vicia orientalis* (Boiss.) Bég. & Diratz (75%), *Lathyrus cicera* L. (50%), and *Pisum sativum* L. (69%). However, there are also some from the scrub and forest: *Carex* sp. (50%), *Juniperus* cf. *excelsa* M.Bieb. (one seed), and *Pyrus syriaca* Boiss. (one seed) (Figure 8).

### 3.2.2. Aramaic

The subsequent Aramean occupation apparently respected the sacred tradition of the place, with the construction of a building composed of three parallel naves; the central one had a religious function (offering table) and the lateral ones (large storage jars) were warehouses. This construction would be closed on at least two of its sides by a porticoed patio and towards the interior and cobbled; it would be over pits made for the extraction of earth in order to make adobe bricks. These pits would be filled with waste [12]. This occupation ended abruptly with a fire and looting, as can be deduced from the fact that all the ceramics in the naves were found broken near the door. The level of fire has provided trunks 10 cm in diameter, innumerable branches of 1 to 2 cm, and the remains of a mat of vegetable fibers that seem to come from the roofs: the wooden beams were placed on the upper part of the walls to cover the width of the construction; on, these the mat was placed to completely close the space and once this operation was carried out, the mat was covered with branches and earth to have the roof completed [15].

Only two samples were analyzed for the Aramean period in Tell Khamîs. *Cyperus* sp. (100%) is the only taxon exclusive or relevant to the Aramean levels from among the five identified. This level, looted and burnt, was completely razed to the ground and affected by countless pits from later periods. As far as we know, it is a temple with three naves (the side naves were used as storerooms) in the center of a courtyard delimited by a gallery open to the interior. Associated with the buildings were a series of irregular pits used for making adobe bricks and later used as rubbish dumps. A $^{14}$C sample gave a calibrated date of 890–758 BC (79.2%) [2], which fits well with the proposed chronology based on the materials and artifacts identified: the ninth century BC.

### 3.2.3. Assyrian

Immediately after the burning of the temple from the Aramean period, the site became a small center of agricultural production, possibly forming part of the belt of farms that surrounded Kar Salmanasar (Tell Ahmar). The abundance of homes and bread ovens stands out, as well as the existence of a set of pits (irregular), corresponding to the first moment of occupation; these were made both to extract earth to make adobe bricks and to transform livestock excrement into fuel (circular) [58]. The Assyrian levels yielded 22 samples that belonged largely to pits and, to a lesser extent, to Tannûr (a fire pot, furnace or oven).

The Assyrian levels produced, as exclusive or relevant, 26 of the 58 taxa identified from this period. Most are segetal weeds: *Heliotropium* sp. (100%), *Ochthodium aegyptiacum* (L.) DC. (100%), *Sinapis* sp. (100%), *Bunias* cf. *erucago* L. (100%), *Alyssum desertorum* Stapf

(100%), *Euphorbia* sp. (100%), *Medicago* sp. (100%), *Trigonella caelesyriaca* Boiss. (100%), *Alcea* sp. (100%), *Reseda luteola* L. (100%), *Rosa canina* L. (100%), *Peganum harmala* L. (100%), *Lolium* sp. (96%), *Chenopodium album* L. (76%), *Medicago astroites* (Fisch. & C.A.Mey.) Trautv. (=*Trigonella astroites* Fisch. & C.A.Mey.) (66%), *Centaurea solstitialis* L. (66%), *Arnebia linearifolia* DC. (65%), *Buglossoides arvensis* (L.) I.M.Johnst. (50%), *Melilotus* sp. (50%), *Polygonum arenarium* Waldst. & Kit. (=*Polygonum venantianum* Clem.) (50%), and *Scorpiurus muricatus* L. (50%). However, some are crops such as *Triticum monococcum* L. (100%) or *Sesamum indicum* L. (92%). Finally, only one grows in scrub and rock crevices: *Capparis zoharyi* Inocencio, D.Rivera, Obón & Alcaraz (88%). *Capparis zohary* has recently been described [59] but its current range is relatively large, extending from south-eastern Spain to the Negev and Euphrates, so it is likely that some of the *Capparis* mentioned in Near Eastern sites correspond to this species [52,60].

The Assyrian period saw a radical change in the characterization of the Tell. Two dwellings were built on the eastern side of the Tell, together with warehouses, a workshop, and a sheepfold. The area changed from one of worship to one of agriculture and livestock farming, with a strong emphasis on livestock. There are central hearths in the dwellings and bread ovens, and in the unoccupied area of the site, there are pits for extracting earth to make adobe bricks, and other smaller, cylindrical pits used to store excrement for later transformation into fuel. Over time, the entire tell was occupied with an anarchic growth, with hearths being placed in the corners of the rooms and collective spaces being created for baking bread. There is no evidence of silos. The settlement was abandoned at a vague point in the seventh century BC.

### 3.2.4. Later Periods

The Assyrian habitat was gradually depopulated to be occupied again, and partially during the phase of Persian rule. The first occupants of this period found a large part of the previous constructions in ruins, although they managed to take advantage of the best-preserved structures for the village. It continued to be, as in the previous phase, an occupation of an agricultural nature that, over time, managed to occupy the entire surface of the Tell. Regarding plant remains. It is worth mentioning a set of small circular silos excavated in the center of the hill, and therefore, well protected by the houses that surround them, and a large circular silo 2.5 m in diameter and 3 m deep, was built with adobe. Apparently, it was protected or inside a fence or a covered structure, of which only a masonry wall with a hinge at one end has been preserved in very poor condition. A cereal deposit with the same characteristics, although 5 m in diameter, appeared in Level 5 of Tell Sera, in the Negev [61]. It is dated to the Persian period, during the fifth and fourth centuries BC. Inside, abundant Attic ceramics and bone spatulas were found. In Israel, these types of silos are common during the Achaemenid domination [62].

This level continued after the Macedonian conquest, although the Tell remained uninhabited during an imprecise period. It is possible that during this time, the settlement began on the plain that surrounds the hill; coinciding with this, we witness the use of the site as a necropolis, as if it were a great burial mound in which to deposit the bodies of important figures in the community, such as can be deduced both from the trousseau and from the sarcophagus in one of the tombs. The finding of a funerary banquet as part of the burial ritual stands out.

The samples collected in 1994 vary between the Hellenistic, Persian and Assyrian chronological horizons, being the best represented set with 29 samples. Finally, 15 samples belong to the Hellenistic or Persian levels. These periods yielded, as exclusive or relevant, 37 taxa among the 49 recovered from the Assyrian–Hellenistic–Persian levels, and of the 38 from the Hellenistic–Persian, most of them are segetal weeds: *Onopordum* sp. (100%), *Euphorbia* sp. (100%), *Teucrium* sp. (100%), *Onobrychis* sp. (100%), *Medicago radiata* L. (100%), *Echinochloa* sp. (100%), *Sherardia arvensis* L. (100%), *Veronica* cf. *syriaca* Roem. & Schult. (100%), *Viola* cf. *pentadactyla* Fenzl (100%), *Eremopyrum* cf. *bonaepartis* (Spreng.) Nevski (94%), *Gypsophila vaccaria* (L.) Sm. (=*Vaccaria pyramidata* Medik.) (93%), *Lepidium*

sp. (83%), *Amaranthus* sp. (68%), *Hyoscyamus aureus* L. (66%), *Ajuga* sp. (66%), *Setaria* sp (66%), *Valerianella coronata* (L.) DC. (57%), *Thymelaea passerina* (L.) Coss. & Germ. (57%), *Trigonella coerulescens* (M.Bieb.) Halácsy (53%), *Adonis aleppica* Boiss. (53%), *Scorpiurus muricatus* L. (50%), *Rumex pulcher* L. (50%), *Hordeum spontaneum* K.Koch (50%), *Carex divisa* Huds. (50%), *Iris aucheri* (Baker) Sealy (50%), and *Panicum miliaceum* L. (50%). The relevant crops are *Lathyrus cicera* L. (50%), *Vitis vinifera* L. (50%), *Triticum aestivum* L. (100%), *Secale cereale* L. (54%), and *Beta vulgaris* L. (100%). Other crops of interest exclusive to this period are *Triticum turgidum* subsp. *dicoccum* (Schrank ex Schübl.) Thell., *Sorghum bicolor* (L.) Moench, *Linum usitatissimum* L., and *Morus alba* L. However, unfortunately, they are represented by a single remnant (seed or fruit), which makes it impossible to assess their importance.

The remarkable proportion of taxa unique to these more recent periods is noteworthy, indicating a marked change in floristic composition from the Late Assyrian period onwards. The first presence of the Persian period (Level 7) took advantage of a large part of the abandoned Assyrian buildings. Agricultural activity continued until the second century BC, in the Hellenistic period. Although this level appears to be clearly Persian, differentiating it from the next (Persian–Hellenistic Level 6) through the dating of the ceramics, there are no significant structural changes. Between the Persian chronology of the site and its abandonment, life continued uninterrupted. In these phases (Levels VII-II) there was a profusion of silos, exceeding fifty in number. It is not easy to assign them to one phase or another due to their state of preservation and because the excavated material corresponds to the abandonment of the silos and their subsequent filling. Therefore, we do not know the exact time at which they were excavated, how long they have been in existence, or what the filling process was.

### 3.3. Habitats of the Identified Taxa

From the fourteen habitat types, two well-differentiated and characterized groups emerge, neatly represented: the clearly anthropogenic ones (cultivated areas (crops); cultivated areas (weeds); disturbed habitats; and nutrient-rich soils, ruderal), which show a constant increase in relevance throughout the time series (Figure 9); and the thickets and natural scrublands (batha, phrygana, shrub-steppes), with a significant presence throughout the record, although decreasing in the most recent levels (Figure 9), likely due to the decrease in their surface area as a result of increasing aridity (Table 4) and overexploitation.

Desert is a habitat present throughout the time series, although in slightly lower proportions in later periods. It is worth noting the presence of saline habitat plants only in the most recent levels, although in low proportions (Figure 9), this could be correlated with the occurrence of a salinization process in the area [2] associated with the increasing aridity (Table 4). Other habitats such as sand, hard rock outcrops, and wet habitats appear in small proportions along the different periods. Wetland-type habitats are represented in minor proportions, but significantly less in more recent periods, which may be related to their transformation into cultivated areas or to increasing aridity (Table 4), or the coincidence of both.

The possible existence of commercial activities, highlighted by the category "Imported" —which includes very remote habitats characteristic of the Mediterranean coast, high mountain, or tropical habitats, during the Middle Bronze and Assyrian periods—and by the presence of Mountain steppe forest species, is noteworthy (Figure 9).

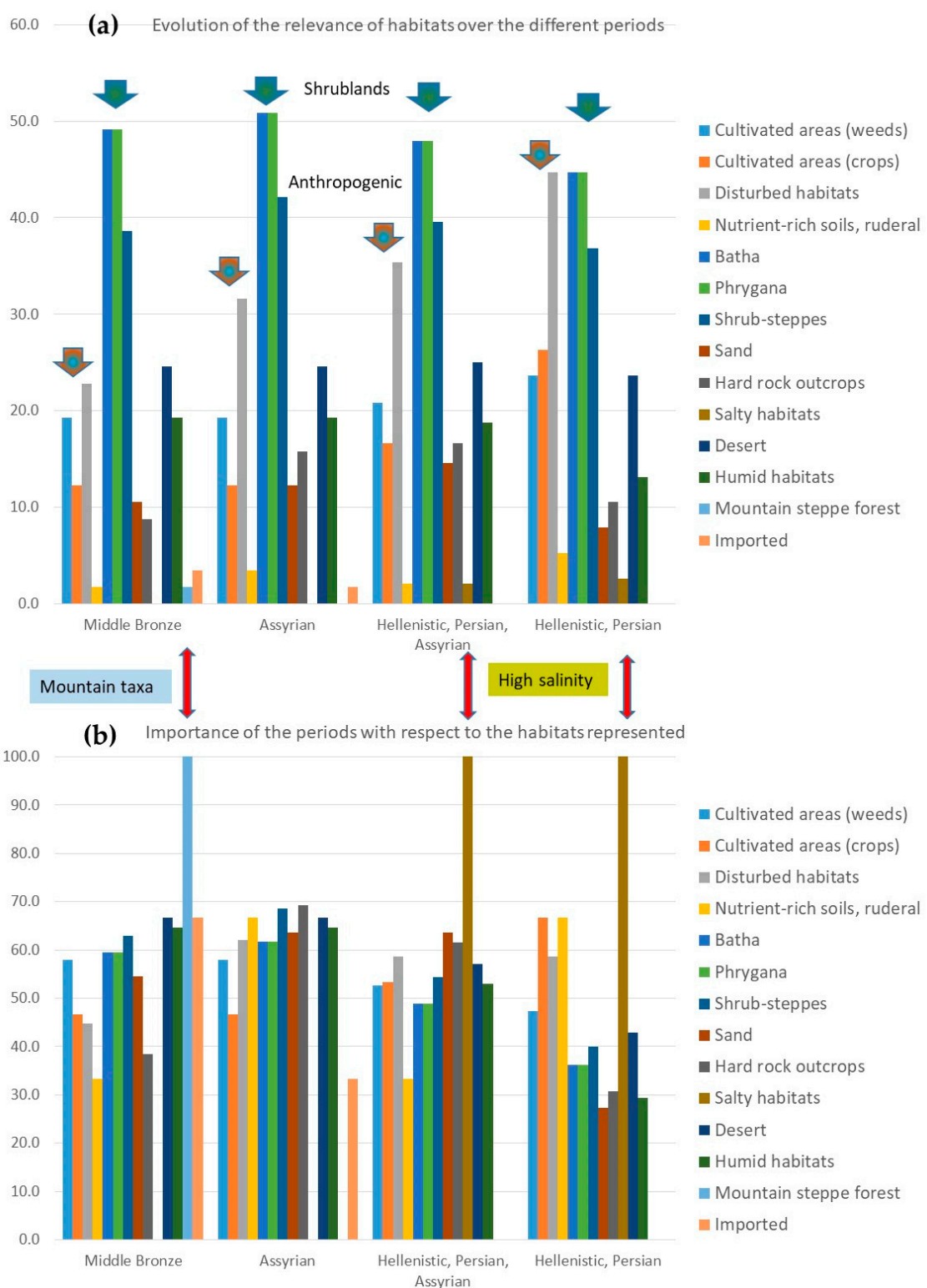

**Figure 9.** Relevance of habitats over the periods analyzed in terms of the plant remains identified from Tell Khamîs: (**a**) Diachronic evolution calculated as the percentage of genera and species growing in each habitat type out of the total number of genera and species identified in each period; (**b**) contribution of each period to the total number of species inhabiting each habitat type, expressed as a percentage: 0 means no occurrence in that period and 100 means that all species of a determinate habitat occured in that period.

**Table 4.** Historical chart of Tell Khamîs and the main environmental events that may have influenced the availability of plant species. Levels, periods, and chronology from Matilla [12]. Paleoclimatology from [63–69].

| Level | Period | Chronology | Climate Events | Refences |
|---|---|---|---|---|
| Khamîs XI | Early Bronze Age | 2800–2500 BC | By around 3000 BC, there was a shift to more enriched $\delta^{18}$O values, similar to the present day. This is suggestive of drier conditions prevailing, but later, it was followed by a relatively high precipitation period. | [10,63,66] |
| Khamîs X | Middle Bronze Age | 2000–1850 BC (ceramic)/1737–1445 BC (14C) | The comparison of the $\Delta$13C values of different crops in different periods confirms increased aridity during the Middle Bronze Age (2000–1600 BC) compared to the later Early Bronze Age (2700–2000 BC), particularly in the north-eastern Syrian territory. In the Mediterranean and Western Asia, the 2200–1900 BC excursion was a period of sudden cooling and increasing aridity, the product of a still-unexplained weakening of North Atlantic Cyclogenesis. It was followed by a relatively high precipitation period. | [10,63,67,68] |
| Khamîs IX | Aramaic period | 9th century BC | Relatively low precipitation. Stable steppe vegetation. High evaporation which, in turn, followed increasing aridity. | [63,65] |
| Khamîs VIII | Assyrian (Neo-Assyrian period) | 8th–7th centuries BC | Late Assyrian Dry Phase c. (700) 650–600 BC | [64] |
| Khamîs VIII-II | Persian–Hellenistic period | 6th–2nd century BC | Later Dry Phase c. 450–350 BC. Dry period based on $\delta^{18}$O levels | [64,66] |

## 4. Discussion

### 4.1. The Place of Tell Khamîs within the Context of Archaeobotanical Evidence from West Asia and Especially the Euphrates Area

The species shared by Tell Khamîs with various sites in the area show maximums in the case of Tell Qara Qûzâq and Tell es-Sweyhat. However, if we consider the relevance of all the different species of each site (dispersion field volume), then Tell Hamman, Ramad, Tell Sabi Abyad, and the ensemble of Troad sites present relatively high values (Table 5).

**Table 5.** Relevant archaeological sites of Syria, Turkey, Iraq, and Iran that present species also identified in Tell Khamîs. Note: values of main habitats represented are in terms of normalized proportion with respect to the maximum. Abbreviations: HABITATS: ANT—Anthropogenic; SHR—Shrubland and other natural habitats; A/N—Proportion ANT/SHR as anthropic index; SAL—Salty habitats; DES—Desert; HU—Humid habitats; IM—Imported. PERIODS: PA—Paleolithic; NE—Neolithic; CH—Chalcolithic; BA—Bronze Age; HI—Hittite; UR—Urartian; IA—Iron Age; As—Assyrian; EL—Elamite; HE—Hellenistic; RO—Roman; SA—Sassanid. SITE PARAMETERS: $s_{sj}$: the shared species-richness values for sites; $R_j$: the dispersion field volume of a site j is the summation of the range sizes of the species occurring at that site; $n_j$: the per-site range size value, expressed as the ratio $R_j/s_j$). CLUSTERS: a to d = based on the number of shared species with Tell Khamîs. * sites with a relatively low number of shared species.

| Country | Sites | Ward | ANT | SHR | A/N | SAL | DES | HU | IM | $s_j$ | $R_j$ | $n_j$ | Periods |
|---|---|---|---|---|---|---|---|---|---|---|---|---|---|
| | | Cluster [1] | | | | Habitats | | | | | Site Parameters | | |
| Syria | Hadidi | a | 33 | 100 | 0.3 | 2 | 15 | 10 | 0 | 27 | 501 | 18.6 | BA-RO |
| Syria | Selenkahiye | a | 34 | 100 | 0.3 | 0 | 16 | 10 | 0 | 37 | 628 | 17.0 | BA |
| Syria | Tell Qara Quzaq | a | 43 | 100 | 0.4 | 1 | 16 | 11 | 2 | 59 | 858 | 14.5 | BA |
| Syria | Tell Khamîs | a | 44 | 100 | 0.4 | 1 | 14 | 11 | 2 | 90 | 1187 | 13.2 | BA-HE |

**Table 5.** *Cont.*

| Country | Sites | Ward | ANT | SHR | A/N | SAL | DES | HU | IM | $s_j$ | $R_j$ | $n_j$ | Periods |
|---------|-------|------|-----|-----|-----|-----|-----|----|----|-------|-------|-------|---------|
| | | Cluster [1] | | | | Habitats | | | | | Site Parameters | | |
| Syria | Hajji Ibrahim | b | 37 | 100 | 0.4 | 0 | 17 | 9 | 0 | 25 | 467 | 18.7 | BA |
| Syria | Tell es Sweyhat | b | 38 | 100 | 0.4 | 0 | 15 | 11 | 0 | 43 | 788 | 18.3 | BA |
| Syria | Tell Hamman et Turkman | b | 39 | 100 | 0.4 | 0 | 17 | 15 | 0 | 38 | 793 | 20.9 | BA |
| Turkey | Asikli Höyük | b | 41 | 100 | 0.4 | 0 | 13 | 10 | 0 | 21 | 481 | 22.9 | NE |
| Syria | Ramad | b | 46 | 100 | 0.5 | 0 | 18 | 14 | 2 | 32 | 704 | 22.0 | NE |
| Syria | Tell Mureybit | b | 47 | 100 | 0.5 | 0 | 22 | 14 | 0 | 32 | 581 | 18.2 | PA |
| Syria | Tell Sabi Abyad | b | 49 | 100 | 0.5 | 0 | 15 | 16 | 0 | 35 | 683 | 19.5 | NE |
| Turkey | Çatalhöyük | b | 52 | 100 | 0.5 | 0 | 22 | 7 | 4 | 20 | 445 | 22.3 | NE |
| Syria | Tell Bouqras | b | 66 | 100 | 0.7 | 0 | 24 | 16 | 0 | 29 | 610 | 21.0 | NE |
| Syria | Tell Abu Hureyra | b | 100 | 78 | 1.3 | 0 | 11 | 17 | 0 | 16 | 376 | 23.5 | PA-NE |
| Turkey | Hacinebi Tepe | c | 37 | 100 | 0.4 | 0 | 17 | 10 | 0 | 30 | 665 | 22.2 | CH-BA |
| Turkey | Çayönü | c | 47 | 100 | 0.5 | 0 | 19 | 12 | 0 | 25 | 552 | 22.1 | NE |
| Turkey | Troad | c | 47 | 100 | 0.5 | 1 | 15 | 13 | 0 | 40 | 689 | 17.2 | BA-IA |
| Turkey | Kurban Höyük | c | 55 | 100 | 0.5 | 0 | 13 | 10 | 0 | 17 | 423 | 24.9 | NE-BA |
| Syria | Ras Shamra | c | 60 | 100 | 0.6 | 0 | 15 | 13 | 0 | 25 | 526 | 21.0 | NE-BA |
| Iran | Deh Luran | c | 60 | 100 | 0.6 | 0 | 23 | 13 | 0 | 20 | 483 | 24.2 | NE |
| Syria | Tell Aswad | c | 70 | 100 | 0.7 | 0 | 21 | 15 | 0 | 26 | 636 | 24.5 | NE |
| Syria | Jerablus Tahtani | c | 72 | 100 | 0.7 | 0 | 20 | 12 | 0 | 17 | 406 | 23.9 | BA |
| Syria | Ghoraife | c | 73 | 100 | 0.7 | 0 | 23 | 14 | 0 | 18 | 444 | 24.7 | NE |
| Iraq | Nimrud | c | 75 | 100 | 0.8 | 0 | 13 | 17 | 0 | 18 | 340 | 18.9 | AS-HE |
| Iran | Bastam | c | 100 | 76 | 1.3 | 0 | 14 | 14 | 0 | 17 | 380 | 22.4 | UR-SA |
| Syria | Jerf el-Ahmar | d | 9 | 100 | 0.1 | 0 | 15 | 6 | 0 | 14 | 231 | 16.5 | NE |
| Syria | Tell al-Raqai | d | 19 | 100 | 0.2 | 0 | 19 | 11 | 0 | 21 | 376 | 17.9 | BA |
| Syria | Dja'de | d | 21 | 100 | 0.2 | 0 | 14 | 7 | 2 | 20 | 321 | 16.1 | NE |
| Syria | Umm al-Marra | d | 30 | 100 | 0.3 | 0 | 21 | 14 | 0 | 27 | 447 | 16.6 | BA |
| Syria | Tell Kerma | d | 34 | 100 | 0.3 | 0 | 20 | 9 | 0 | 19 | 369 | 19.4 | BA |
| Syria | Tell Atij | d | 38 | 100 | 0.4 | 0 | 23 | 15 | 0 | 25 | 522 | 20.9 | BA |
| Syria | Qaramel | * | 0 | 100 | 0.0 | 0 | 20 | 7 | 0 | 7 | 96 | 13.7 | NE |
| Syria | Tepecik | * | 25 | 100 | 0.3 | 0 | 10 | 15 | 0 | 9 | 153 | 17.0 | CH-BA |
| Syria | Tell el Kown | * | 33 | 100 | 0.3 | 0 | 29 | 8 | 0 | 12 | 282 | 23.5 | NE |
| Iran | Tappeh Sarafabad | * | 38 | 100 | 0.4 | 0 | 19 | 13 | 0 | 7 | 168 | 24.0 | NE |
| Syria | Tell Schech Hamad | * | 38 | 100 | 0.4 | 0 | 23 | 31 | 0 | 7 | 144 | 20.6 | AS |
| Syria | Umm Qseir | * | 55 | 100 | 0.5 | 0 | 18 | 0 | 0 | 7 | 221 | 31.6 | CH |
| Syria | Tell es Sinn | * | 62 | 100 | 0.6 | 0 | 19 | 19 | 0 | 12 | 270 | 22.5 | NE |
| Iran | Susa | * | 63 | 100 | 0.6 | 0 | 19 | 13 | 0 | 8 | 214 | 26.8 | SA |
| Syria | Tell Aqab | * | 67 | 100 | 0.7 | 0 | 0 | 0 | 0 | 7 | 159 | 22.7 | CH |
| Syria | Tell Brak | * | 69 | 100 | 0.7 | 0 | 6 | 0 | 0 | 11 | 283 | 25.7 | CH |
| Iran | Malyan | * | 73 | 100 | 0.7 | 0 | 18 | 18 | 0 | 14 | 299 | 21.4 | EL |
| Turkey | Hacilar | * | 95 | 100 | 0.9 | 0 | 16 | 16 | 5 | 15 | 374 | 24.9 | NE-CH |
| Turkey | Erbaba | * | 100 | 73 | 1.4 | 0 | 9 | 0 | 0 | 10 | 277 | 27.7 | NE |
| Turkey | Beycesultan | * | 100 | 94 | 1.1 | 0 | 13 | 13 | 0 | 10 | 243 | 24.3 | NE-BA |
| Turkey | Can Hasan | * | 100 | 84 | 1.2 | 0 | 11 | 11 | 5 | 14 | 377 | 26.9 | NE |
| Turkey | Ilipinar | * | 100 | 87 | 1.2 | 0 | 13 | 7 | 0 | 10 | 318 | 31.8 | NE |
| Turkey | Girikihaciyan | * | 100 | 77 | 1.3 | 0 | 8 | 8 | 0 | 9 | 294 | 32.7 | CH |
| Turkey | Bogazköy (Hattusa) | * | 100 | 0 | - | 0 | 0 | 0 | 0 | 5 | 152 | 30.4 | HI |
| Turkey | Milet | * | 100 | 70 | 1.4 | 10 | 0 | 0 | 0 | 6 | 139 | 23.2 | IA |
| Syria | Tell Halula | * | 100 | 93 | 1.1 | 0 | 7 | 7 | 0 | 10 | 262 | 26.2 | NE |
| Syria | Korucutepe | * | 100 | 92 | 1.1 | 0 | 15 | 15 | 0 | 8 | 184 | 23.0 | BA |
| Iran | Ali Kosh | * | 100 | 17 | 6.0 | 0 | 17 | 17 | 0 | 5 | 159 | 31.8 | NE |
| Iran | Tepe Hissar | * | 100 | 50 | 2.0 | 0 | 17 | 0 | 0 | 8 | 252 | 31.5 | BA |

**Table 5.** *Cont.*

| Country | Sites | Ward | ANT | SHR | A/N | SAL | DES | HU | IM | $s_j$ | $R_j$ | $n_j$ | Periods |
|---------|-------|------|-----|-----|-----|-----|-----|----|----|-------|-------|-------|---------|
| | | **Cluster** [1] | | | | **Habitats** | | | | | **Site Parameters** | | |
| Iraq | Tell Magzalija | * | 100 | 80 | 1.3 | 0 | 20 | 10 | 0 | 7 | 220 | 31.4 | NE |
| Iraq | Umm Dabaghiyah | * | 100 | 38 | 2.7 | 0 | 0 | 0 | 0 | 6 | 180 | 30.0 | NE |
| Iraq | Jarmo | * | 100 | 71 | 1.4 | 0 | 0 | 0 | 0 | 5 | 168 | 33.6 | NE |
| Iraq | Choga Mami | * | 100 | 89 | 1.1 | 0 | 16 | 11 | 0 | 15 | 435 | 29.0 | CH |
| Iraq | Tell Chragh | * | 100 | 50 | 2.0 | 0 | 13 | 13 | 0 | 5 | 176 | 35.2 | CH |
| Iraq | Tell Es-Sawwan | * | 100 | 0 | - | 0 | 0 | 0 | 0 | 7 | 240 | 34.3 | CH |
| Iraq | Tell Taya | * | 100 | 50 | 2.0 | 0 | 0 | 10 | 0 | 6 | 172 | 28.7 | BA |
| Iraq | Ur | * | 100 | 20 | 5.0 | 0 | 0 | 0 | 0 | 5 | 167 | 33.4 | BA-AS |
| Iraq | Tell ed-Der | * | 100 | 100 | 1.0 | 0 | 33 | 33 | 0 | 9 | 212 | 23.6 | BA |
| Iraq | Tell Karrana | * | 100 | 89 | 1.1 | 0 | 11 | 6 | 0 | 14 | 403 | 28.8 | BA |
| Iraq | Tell Yelkhi | * | 100 | 0 | - | 0 | 0 | 0 | 0 | 7 | 236 | 33.7 | BA |
| Iraq | Mahmudiya | * | 100 | 56 | 1.8 | 0 | 11 | 22 | 0 | 8 | 229 | 28.6 | HE |

[1] Note: the clusters, a to d, in this table are based on presence absence of species and differ from those in Figure 10 that are calculated on the base of main habitats of the species.

The Ward's tree based on the list of species shows four distinct clusters of sites (a–d in Table 5); the factorial analysis (PCoA) gives similar results. The Ward's tree (Figure 10) based on the habitats of species shows only two distinct clusters of sites: one around Tell Khamîs characterized by a major proportion of natural habitats, and the other group characterized by the prevalence of anthropogenic habitats as a function of the shared species, in addition to a few sites not well differentiated, characterized by similar proportions of anthropogenic and natural habitats (Figure 10).

The best-characterized group, (cluster a, Table 5), comprises the Syrian sites of Tel Khamîs, Tell Qara Qûzâq, Selenkahiye, and Hadidi, with 39 species occurring in at least three of the four sites (75%). Species exclusive or almost exclusive to this group are *Bupleurum subovatum* Link ex Spreng., *Valerianella vesicaria* Moench, and *Glaucium aleppicum* Boiss. & Hausskn. ex Boiss., inhabiting batha, phrygana and shrub-steppes. Qara Qûzâq is only a few kilometers away from Khamîs, and it should be expected that the two sites share a large number of taxa (wild and cultivated) and profile of habitat types. The fundamental difference is that Qara Qûzâq is next to the Euphrates and Khamîs is 5 km away, with rain-fed agriculture being practiced in the immediate vicinity. In any case, in addition to their geographical proximity, they share Middle Bronze Age and Early Bronze Age levels. Selenkahiye and Hadidi are also located along the Euphrates, 57 km and 40 km from Khamîs, respectively. In general, this group (cluster a) has, like cluster d, the lowest proportions of anthropogenic habitats (Table 5).

The second group (cluster b, Table 5) includes ten sites, of which eight are in Syria (Tell es Sweyhat, Tell Hamman et Turkman, Tell Sabi Abyad, Ramad, Tell Mureybit, Tell Bouqras, Hajji Ibrahim, and Tell Abu Hureyra) and only two in Turkey (Asikli Höyük and Çatalhöyük). This group of sites is characterized by nine species occurring, at least eight of which, evidently, are also present in Tell Khamîs (*Hordeum spontaneum* K.Koch, *Triticum monococcum* L., *Triticum turgidum* subsp. *dicoccum* (Schrank ex Schübl.) Thell., *Vicia ervilia* (L.) Willd., *Pisum* sp., *Silene* sp., and *Bolboschoenus maritimus* (L.) Palla (=*Scirpus maritimus* L.)). Most of these sites share predominantly, with Tell Khamis, species characteristic of natural habitats; the exceptions are Tell Bouqras, which has similar proportions of natural and anthropogenic habitats represented, and Tell Abu Hureyra (Table 5), where anthropogenic habitats predominate, perhaps due to the focus set by researchers on the relevance of early agriculture. This group includes sites in the Euphrates valley, but with different chronologies (Table 5, Tell es Sweyhat (incl Hajji Ibrahim), Tell Mureybit, Tell Abu Hureyra, and Tell Bouqras). Settlements in the Balikh valley, a tributary of the Euphrates, are also present with different sequences (Hamman et Turkman, Tell Sabi Abyad). Finally, Ramad, 20 km south of Damascus, has little to do with Khamîs from a geographical or

chronological point of view. The most pronounced difference between the two Turkish sites of cluster b, with respect to the first group that included Tell Khamîs, is the chronological horizon; both belong to the Neolithic period, in a range between the Pre-Pottery Neolithic (8000–7500 BC) and the Early Ceramic Neolithic (6200–5500 BC).

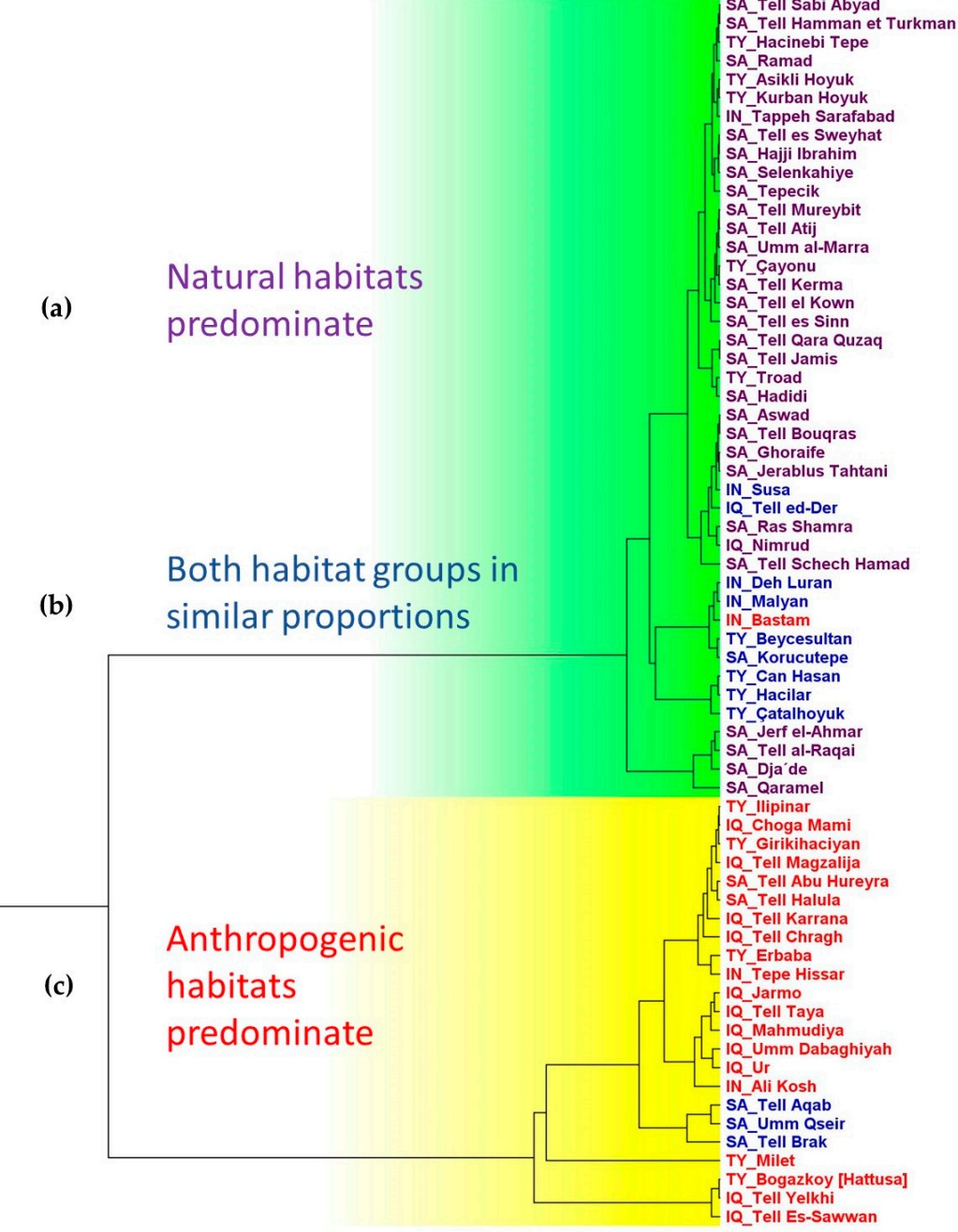

**Figure 10.** Main groups of sites resulting of the chi-square analysis of dissimilarities based on the habitats of the species identified in Tell Khamîs and shared with the different sites: (**a**) main group around Tell Khamîs characterized by the large set of natural habitats; (**b**) not well-differentiated group characterized by similar proportions of anthropogenic and natural habitats; (**c**) group characterized by the prevalence of anthropogenic habitats [1]. Colour codes for characters: Red, anthropogenic habitats. Blue: intermediate. Purple: natural predominate. ([1] Note: the clusters, a to c, in this figure are based the main habitats of the species and differ from those in Table 5 that are exclusively calculated on presence absence of species).

The third group (cluster c, Table 5) is the most geographically heterogeneous as it includes eleven sites from Syria (Aswad, Ras Shamra, Ghoraife, and Jerablus Tahtani), Turkey (Troad sites, Hacinebi Tepe, Çayönü, and Kurban Höyük), Iraq (Nimrud) and Iran (Deh Luran and Bastam). All these sites share, with Tell Khamîs, species characteristic of natural habitats; however, anthropogenic habitats are represented in relatively higher proportions than in the previous ones, with an extreme in Bastam where anthropogenic habitats predominate (cluster c in Table 5 and group b in Figure 10). This group of sites is characterized by six species, or at least five, of which all occur in Tell Khamîs (*Lolium* sp., *Triticum aestivum* L./*T. turgidum* L. subsp. *durum* (Desf.) van Slageren, *Triticum monococcum* L., *Triticum turgidum* subsp. *dicoccum* (Schrank ex Schübl.) Thell., *Lens culinaris* Medik., and *Linum usitatissimum* L.). These are all crops or crop weeds characteristic of anthropogenic habitats.

Of the Syrian sites, Jerablus Tahtani is on the Euphrates; Ras Shamra is on the Mediterranean coast; Tell Aswad is about 50 km from Damascus, along a tributary of the Barada; and Ghoraife, a Neolithic site, is also located near Damascus. The shared characteristic in the third group between the Turkish, Iraqi, and Iranian sites seems to be the geographical situation and the climate, with a relevance of natural scrubland and steppe. The Turkish sites of Hacinebi Tepe, Çayönü, and Kurban Höyük are located in the southeast of the country, in a region presently dominated by oak forest. The Nimrud and Bastam deposits do not seem to be related to each other or to the Turkish deposits, but they are all located at almost the same latitude, and, except for the Iraqi deposit, the rest are above 830 m of altitude above sea level.

The fourth group (cluster d, Table 5) is geographically homogeneous as it includes six Syrian sites (Umm al-Marra, Tell Atikh, Dja'de, Tell al-Raqai, Tell Kerma, and Jerf el-Ahmar). This group of sites is characterized by seven species that occur in at least five of them and in Tell Khamîs (*Bellevalia* sp., *Gypsophila vaccaria* (L.) Sm. (=*Vaccaria pyramidata* Medik.), *Silene* sp. (incl. *S. vivianii* Steud. and *S. vulgaris* (Moench) Garcke), *Hordeum spontaneum* K.Koch, *Eremopyrum* cf. *bonaepartis* (Spreng.) Nevski, *Medicago astroites* (Fisch. & C.A.Mey.) Trautv. (=*Trigonella astroites* Fisch. & C.A.Mey.), and *Medicago radiata* L. (=*Trigonella radiata* (L.) Kuntze)). All these sites share, with Tell Khamîs, mostly species characteristic of natural habitats, with a very low representation of anthropogenic habitats (they appear embedded in the large cluster a of Figure 10). Umm al-Marra is located between the Euphrates and Aleppo; Tell Atikh in the Khabur valley has an Early-Middle Bronze horizon; Dja'de el-Mughara is Neolithic and is located between Tell Khamîs and Tell Qara Qûzâq, along the Euphrates, as is the Neolithic Jerf el-Ahmar; Tell al-Raqai is located in the Khabur area; and on the same river is Tell Kerma.

When taking into account the proportion in percentage of the different habitats in the species lists, the number of analyzable sites increases; Figure 10 shows the existence of a large group of sites, mostly located in Iraq, which mainly share, with Tell Khamîs, species characteristic of anthropogenic habitats, especially cultivated species and weeds. It cannot be ruled out that this may be due to a commercial relationship with present-day Iraq and the importation of foodstuffs from there. However, if we consider that in this case, the number of shared species is much smaller—less than fifteen in general (Table 5)—and most of them are typical of anthropogenic habitats, notably crops, it is much more likely that the climatic and biogeographical differences led the natural vegetation to completely differ and that the synanthropic species represented a certain type of globalization. In other words, the crops (barley and other cereals and legumes) and associated weeds spread over very different areas, beyond the natural bioclimatic frontiers.

### 4.2. The Role of Plants: Crops and Weeds

The site has undergone a temporal evolution with a maximum of documented activity in the Middle Bronze Age period, where a 4 m$^3$ silo filled with barley—equivalent to $52 \times 10^6$ grains of barley, of which only 1/1000th has been recovered—together with the associated room, has yielded about 10,000 remains. It is worth mentioning that the Middle

Bronze Age in the Euphrates area is characterized by an increase in aridity and a decrease in temperatures (Table 4).

Boraginaceae has contributed thousands of fruits with mineralized walls of the genera *Arnebia, Buglossoides*, and *Heliotropium*, with a maximum in the Assyrian period and a significantly smaller proportion in the Middle Bronze period.

Despite the low density of plant remains found at Tell Khamîs compared to other similar sites, our approach, in this article, reveals expressions of land use and vegetation cover throughout all the chronological sequences analyzed, noting the use of plant species and their context within the archaeological record. The considered "founding crops" for Neolithic agriculture such as lentil (*Lens culinaris*), bitter vetch (*Vicia ervilia*), barley (*Hordeum vulgare*), and wheat (*Triticum aestivum*), among others [1], seem to have a continuous dynamic within the plants of economic use in Tell Khamîs. In spite of the long sequence of stages within the site, the changes in these food plants are hardly noticeable.

Barley is one of the most important and common taxa in Tell Khamîs in all the samples analyzed, both in ubiquity and absolute count, appearing abundantly in all the periods sampled. Compared to wheat, barley is more resistant to drought; its greater resistance to soil salinity is notable [2], becoming the most reliable crop with the onset of aridity [64,68,69]. During the Bronze Age, we find similar proportions of barley in almost all Syrian sites, demonstrating intensive production as a common feature. In the modern Lake Assad area, this proportion of barley is much higher during the Bronze Age, which could indicate a special perspective towards this cereal [4]. Other studies point towards the utilization of barley grain as animal fodder, which would allow herders to feed a large number of herd animals to exchange the surplus [70].

As opposed to barley, wheat, during the Bronze Age in Syria, had decreasing economic importance, as shown by the low proportion in absolute counts analyzed at the sites [71]. In turn, it seems that pulses such as lentils and bitter vetch, with a certain tolerance to water stress, complemented the diet of the Tell Khamîs population. Although it is true that their cultivation is widespread in the Near East where these are present at a large number of sites [4], the total seed count in Tell Khamîs is significantly low, which could be related to a locally lesser relevance or, simply, to taphonomic processes that particularly degraded these types of seeds. This is reflected in the poor conservation state of the pulse seeds (Figures 6 and 7).

Recent research suggests that the growing populations of northern Mesopotamia during the Bronze Age were unable to feed themselves from the food catchment areas in the surroundings of cities. Evidence has been found of the practice of procurement, storage, and redistribution of agricultural products, organized from a central administration. This fact is reinforced by the architectural evidence itself: at Tell Khamîs, we find a silo of large proportions associated with a space that presumably had the function of a temple. At other sites such as Tell Brak, Tell Leilan, Tell al-Raqa'I, and Tell Kerma, large storage structures have also been excavated, which could reflect the presence of a central power and the importance of a complex economic organization with movements of agricultural products at this time [72]. Textual references also provide us with good evidence of temple property acquisition and storage practices. Bronze Age texts show how barley functioned as payment for goods and services [73]. Temples, considered the abode of the gods [74], were used as storage for agricultural products that farmers delivered for redistribution as payment, or rations among the staff and artists or craftsmen who worked full time.

### 4.3. Significance and Profile of the Remains from Characteristic Contexts and Structures

With the exception of those from silo contents, and from rooms used as kitchens or pantries for food storage, the rest of the samples present, as aforesaid, a high proportion of Boraginaceae fruits (*Arnebia decumbens* Coss. & Kralik, *Arnebia linearifolia* DC., *Buglossoides arvensis* (L.) I.M.Johnst., and *Buglossoides tenuiflora* (L.f.) I.M.Johnst.); these are very resistant due to the natural mineralization of their walls, suggesting that they were abundant throughout the different periods and have been preserved in the strata.

Two infant graves from the Early Bronze Age contain small amounts of barley Hordeum vulgare L. var. vulgare, in a relatively high proportion, associated with between two and nine species of weeds, especially *Buglossoides tenuiflora* (L.f.) I.M.Johnst. and *Arnebia linearifolia* DC. (079 and 080/2017). It is not excluded that barley and weeds entered the level at different times, the former during burial and the rest during the gradual filling of the pit [75].

From the large Middle Bronze silo, sample 061/2017, with 4963 whole remains composed almost exclusively of *Hordeum vulgare* L. var. *vulgare*, was recovered. Particularly relevant is the Middle Bronze room that gives access to the large silo and that has also provided a large number of remains (4365) in sample 105/2017, which is almost exclusively composed of *Hordeum vulgare* L. var. *vulgare*.

The basal area of a reused silo (Sample 096/2017) offered remains of its earlier contents from the Assyrian period, consisting mainly of *Buglossoides tenuiflora* (L.f.) I.M.Johnst. and a few barley fruits.

The pits are holes that, having been originally intended for various uses, such as silos, were subsequently reused and filled with remains in later periods, likely after their amortization but possibly simultaneously [76]. Their content, in terms of diversity and abundance of plant remains, is very variable. In them, we find domestic species of habitual consumption such as barley together with a very varied repertoire of herbs and other species such as *Peganum harmala*, which may have had medicinal or ritual uses. In the nearby Tell Qara Qûzâq the silos of MB II and EB III-IV show similar compositions [20].

The samples of this type that provided the most remains of fruits and seeds are the following: From the Middle Bronze Age, sample 075/2017, with 247 whole remains, contains ten species, among which are *Buglossoides tenuiflora* (L.f.) I.M.Johnst., and *Hordeum spontaneum* K.Koch; and sample 071/2017, with 231 whole remains, predominantly of *Hordeum vulgare* L. var. *vulgare*, but also *Buglossoides tenuiflora* (L.f.) I.M.Johnst. and nine other species. From the Assyrian period, it is worth mentioning sample 087/2017, with 1073 whole remains belonging to 16 species, with 24% being barley (*Hordeum vulgare* L. var. *vulgare*), and other relevant species such as *Heliotropium* sp., *Buglossoides arvensis* (L.) I.M.Johnst, and *Peganum harmala* L. Sample 092/2017, also from Assyrian levels, yielded 261 entire remains belonging to 15 species, especially *Lolium* sp., *Hordeum vulgare* L. var. *vulgare*, and *Buglossoides tenuiflora* (L.f.) I.M.Johnst.

Middle Bronze age rooms provided abundant remains of seeds and fruits, especially the kitchens, pantries and those that gave access to the silos. Notable samples from kitchens are: 073/2017, with 4777 whole remains, of which 99% are barley (*Hordeum vulgare* L. var. *vulgare*) but which also contains some remains of 11 other species; and 076/2017, with 393 whole remains, composed mainly of *Hordeum vulgare* L. var. *vulgare*, but also weeds such as *Taeniatherum* sp., *Galium aparine* L. s.l., and 8 other species.

The Middle Bronze food pantries provided 1293 whole remains in sample 078/2017, and 572 in sample 077/2017, with a predominance of *Hordeum vulgare* L. var. vulgare and *Vicia ervilia* (L.) Willd.

The fillings of the abandoned Assyrian bread ovens (*tannûr*) offer few remains, especially of *Buglossoides tenuiflora* (L.f.) I.M.Johnst. (Sample 063/2017) or *Bromus sterilis* L. (Sample 086/2017). These are possibly the remains of combustion during the last use.

Considering the abundance of plant remains and the better preservation of the different excavated areas and levels, the most interesting periods analyzed, from an archaeobotanical point of view, at Tell Khamîs range from the Middle Bronze Age to the Assyrian. More than half of the taxa identified in the Bronze Age levels are exclusive or almost exclusive to the Bronze Age period and, thus, differ from the list of taxa recovered from the sediments of later periods, including Aramaic and Assyrian. The later periods, especially Hellenistic, together with some of the upper horizons of the Assyrian strata, are very disturbed, and although numerous plant remains have been recovered from them, the archaeobotanical information could be less interesting from an archaeological point of view. However, there are a significant number of taxa exclusive to the more recent periods. This suggests

a marked change in floristic composition from the Late Assyrian period ahead. This drastic shift in floristic composition could be due to profound environmental alterations influenced by human activity, but also, especially, to the climatic change that led to greater aridity (Table 5).

## 5. Conclusions

The site has undergone a temporal evolution with a maximum of documented activity in the Middle Bronze Age period, where a 4 m$^3$ silo filled with barley, equivalent to $52 \times 10^6$ grains of barley, of which only 1/1000th has been recovered, is worth mentioning. Boraginaceae have contributed thousands of fruits with mineralized walls of the genera *Arnebia*, *Buglossoides*, and *Heliotropium*, with a maximum in the Assyrian period and a significantly smaller proportion in the Middle Bronze period.

Unfortunately, the most interesting taxa from a paleoenvironmental and cultural point of view are represented by only one or very few seeds, which prevents us from progressing in the more detailed reconstruction of the resources used and the environmental impact of the activities associated with the site of Tell Khamîs.

The remarkable proportion of taxa unique to the more recent periods is noteworthy, indicating a marked change in floristic composition from the Late Assyrian period onwards. In parallel, it is worth noting that more than half of the taxa identified in Bronze Age levels are unique or almost unique to the period and, thus, mark its differentiation from later periods, including the Aramaic and Assyrian.

Of the thirteen habitat types, plus the category of unlikely habitats for the area, two well-differentiated and -characterized groups emerge, clearly represented: the clearly anthropogenic habitats, which show a constant increase in relevance throughout the time series, and the natural scrublands and thickets, with a significant presence throughout the record, albeit decreasing in the most recent levels. It is worth noting the presence of saline habitats only in the most recent levels, which could be correlated with the occurrence of a salinization process in the area. Humid-type habitats are represented in smaller proportions, but significantly less in the most recent periods; this may be related to their transformation into cultivated areas, to the increase in aridity, or to both.

**Supplementary Materials:** The following supporting information can be downloaded at: https://www.mdpi.com/article/10.3390/heritage5030088/s1, Table S1: Numbers of remains (entire and fragments) for the different taxa and samples; Table S2: Descriptions of the series of comparison sites from the Near East; Table S3: Binary presence–absence matrix of shared species with a series of sites from the Near East.

**Author Contributions:** Conceptualization, J.V. and D.R.; methodology, D.R. and J.V.; software, D.R and J.V.; validation, J.V., G.M.-S., C.O. and D.R.; formal analysis, D.R.; investigation, J.V.; resources, G.M.-S.; data curation, J.V, D.R. and C.O..; writing—original draft preparation, D.R. and J.V.; writing—review and editing, G.M, J.V., D.R.; visualization, D.R., J.V. and G.M.-S.; supervision, C.O.; project administration, G.M.-S.; funding acquisition, G.M.-S. All authors have read and agreed to the published version of the manuscript.

**Funding:** This research received no external funding.

**Institutional Review Board Statement:** Not applicable.

**Informed Consent Statement:** Not applicable.

**Data Availability Statement:** Not applicable.

**Acknowledgments:** We deeply acknowledge Lucía Cazorla Blázquez for the translation and linguistic revision of this document, and Maria-Teresa Coronado for the realization of carbonized seed images from FE-SEM. We offer thanks to the Embassy of Spain in Syria for the support provided while we were excavating; to the General Directorate of Antiquities and Museums of the Arab Republic of Syria both for having allowed us to excavate in Tell Khamîs and for the support provided to the botanical expedition carried out in 1999; to Dina Bahkur, who accompanied us in the different excavation campaigns; to Abbas al-Hamza and his entire extended family from Qara Quzaq; and to the members of the Archaeological Mission of the University of Murcia in Qara Quzaq (Syria). This article is part of the PhD thesis research of one of the authors, Javier Valera.

**Conflicts of Interest:** The authors declare no conflict of interest.

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
