# Peer review of "Archaeobotanical Study of Tell Khamîs (Syria)"

_heritage, doi:10.3390/heritage5030088_

Round 1

Reviewer 1 Report

Dear Authors,

I highly appreciate that you have revised your work according to my suggestions to its initial version. Indeed, the work is promising, and it can be interesting to the international research community. The manuscript is based on a sound research project, and it is well-illustrated. Nonetheless, I have some additional comments:

1)      Introduction: please, do not start it with your objectives. These should be moved to the end of the section.

2)      Section 3: it is a MUST for papers in international journals to separate Results from Discussion. Results = only your direct findings. Discussion = interpretations of your findings, consideration of methodological limitations, and comparisons to the other studies. You can extract this information from the section 3, make some additions, and merge with what is now section 4.

3)      I strongly feel that this manuscript needs a historical chart and may be palaeoenvironmental schemes to better represent your interpretations related to historical changes in the study area.

4)      Please, re-read the revised version to strengthen the logic of storytelling.

Author Response

Responses to reviewer 1:

Dear referee, in the following text we present the responses to your comments and suggestions that we deeply appreciate.

Dear Authors,

I highly appreciate that you have revised your work according to my suggestions to its initial version. Indeed, the work is promising, and it can be interesting to the international research community. The manuscript is based on a sound research project, and it is well-illustrated. Nonetheless, I have some additional comments:

Many thanks for your comments that fostered the improvement of our paper

1)      Introduction: please, do not start it with your objectives. These should be moved to the end of the section.

Done, thanks.

2)      Section 3: it is a MUST for papers in international journals to separate Results from Discussion. Results = only your direct findings. Discussion = interpretations of your findings, consideration of methodological limitations, and comparisons to the other studies. You can extract this information from the section 3, make some additions, and merge with what is now section 4.

Yes we agree, in fact we started this approach in the resubmitted versión, but failed to change the headings. We revised now the structure and contents of both sections following your advise and expanded discussion section with a new subsection folowing the advise of reviewer 2.

3)      I strongly feel that this manuscript needs a historical chart and may be palaeoenvironmental schemes to better represent your interpretations related to historical changes in the study area.

Thanks, We tried to summarize this information in the new Table 4 (thus former Table 4 become Table 5). This table is cited in several places along the manuscipt where appropriate.

4)      Please, re-read the revised version to strengthen the logic of storytelling.

We reorganized some paragraphs and introduced others new in order to improve the storytelling.

Reviewer 2 Report

It seems that your samples are drawn from a variety of diverse contexts-I would like to see some discussion of the impact of these diverse contexts on your conclusions. I would expect a silo, for instance to reflect a very different set of formation processes than a tomb or grave fill, yet you don't give much consideration to the inherent differences in your contexts nor their impact on your interpretation. I think that such a discussion is necessary and will recommend accept after minor revision. Otherwise, a very interesting study.

Author Response

Responses to reviewer 2

Dear referee, in the following text we present the responses to your comments and suggestions that we deeply appreciate.

It seems that your samples are drawn from a variety of diverse contexts-I would like to see some discussion of the impact of these diverse contexts on your conclusions.

I would expect a silo, for instance to reflect a very different set of formation processes than a tomb or grave fill, yet you don't give much consideration to the inherent differences in your contexts nor their impact on your interpretation.

I think that such a discussion is necessary and will recommend accept after minor revision. Otherwise, a very interesting study.

Dear referee, many thanks! Yes we agree with you and trying to be succinct we commented on the main contexts and structures and the seed samples recovered from each type in a new section under: 4.3. Significance and profile of the remains from characteristic contexts and structures

This manuscript is a resubmission of an earlier submission. The following is a list of the peer review reports and author responses from that submission.

Round 1

Reviewer 1 Report

The manuscript submitted by Valera et al. presents new results of a site that is occupied over a long period of time in current Syria. The number of samples analysed per phase is moderate (or low) but there are some valuable finds, such as large charred stores of grain. Its suitability for the journal Heritage is questionable, considering the current focus of the paper (the authors could make this more clear in the text), but the most important aspect is that the work does not really fulfill important standards of current archaeobotanical research.

The authors did a big effort to document seeds with SEM microscopy. Unfortunately, often there seem to be either missidentifications or pictures that do not help the reader to confirm the author's identifications. This is the case of some of the pulses (Lens, Pisum), Secale (the 3 pictures show a Lolium caryopse), and Astragalus (why not Medicago sp.?), but also Galium aparine (the detail picture does not show the typical cell structure) and Fallopia convolvulus (this seems to be Rumex). The authors use old nomenclature for wheats that should be explained, such as Triticum aestivum L. (are you sure durum can be excluded?) and Triticum parvicoccum. Figures must be rearranged per species. Currently pictures of one single speciment appear separated in the plates and even in different figures.

This is, in my opinion, enough to reject the paper (unfortunately).

I must also say though, that the paper has no clear goals, so the reader does not understand why the analysis of the data is as it is (and it is not a common analysis in archaeobotany, so it would require some explanation). The site is not clearly presented with a table that summarizes totals of finds per type of feature and phase. That is the only way to understand the meaning of the data. Archaeobotanical data is not comparable to pollen or other proxies, since it originates from human activities and can only be compared taking them into consideration. Archaeobotanically, the comparison between sites on a large regional scale based on the presence/absence of any species does not make a lot of sense due to difficulties in the identification process and the local flora diversity. This is done with particular questions (i.e. crops grown). What is the question behind this analysis?

Above all, where is the discussion of the data presented? I only see description of results, including the regional comparison. The discussion section is full of paragraphs describing the site in general which should have appeared in the introduction. There is no actual interpretation on the meaning of the data generated.

I consider that the whole project needs more work before aiming for a publication or maybe a complete reorentation of the paper that allows a more meaningful and interesting paper to be published.

There are some minor English mistakes, such as the use of land instead of sediment.

Reviewer 2 Report

Dear Authors,

I've examined your manuscript with a great interest. The information is novel and internationally important. The manuscript is based on a sound research project, and it is well-written and well-illustrated.

  • Introduction: please, start with more general issues – for instance, the importance of archaeobotanical investigations in the Middle East.
  • Introduction: site description and the related figures can be replaced to the next section.
  • Materials and Methods: This section needs a scheme showing the sampling points. Another scheme demonstrating the local stratigraphy (time frame) would also be helpful.
  • Figure 8: methodological notes from its caption can be added to the methodological section.
  • Please, check whether all tools and indices used in the subsection 3.3 are explained in the methodological part of the work.
  • This paper needs a new section, namely Discussion. There, you need to present interpretations of what do your findings mean. Particularly, I recommend to pay attention to the general historical and environmental changes in the entire macroregion.
  • I strongly recommend to cite more works published in the past three-five years. Most probably, this literature can be related to the proposed Discussion.
  • The writing is ok, but I kindly ask you to avoid too short, one-sentence paragraphs.